# How Much Over-parameterization Is Sufficient to Learn Deep ReLU Networks?

**Zixiang Chen**[†,*]  **Yuan Cao**[†,*]  **Difan Zou**[†,*]  **Quanquan Gu**[†]
[†]Department of Computer Science, University of California, Los Angles
`{chenzx19,yuancao,knowzou,qgu}@cs.ucla.edu`

## Abstract

A recent line of research on deep learning focuses on the extremely over-parameterized setting, and shows that when the network width is larger than a high degree polynomial of the training sample size $n$ and the inverse of the target error $\epsilon^{-1}$, deep neural networks learned by (stochastic) gradient descent enjoy nice optimization and generalization guarantees. Very recently, it is shown that under certain margin assumptions on the training data, a polylogarithmic width condition suffices for two-layer ReLU networks to converge and generalize (Ji and Telgarsky, 2020). However, whether deep neural networks can be learned with such a mild over-parameterization is still an open question. In this work, we answer this question affirmatively and establish sharper learning guarantees for deep ReLU networks trained by (stochastic) gradient descent. In specific, under certain assumptions made in previous work, our optimization and generalization guarantees hold with network width polylogarithmic in $n$ and $\epsilon^{-1}$. Our results push the study of over-parameterized deep neural networks towards more practical settings.

## 1 Introduction

Deep neural networks have become one of the most important and prevalent machine learning models due to their remarkable power in many real-world applications. However, the success of deep learning has not been well-explained in theory. It remains mysterious why standard optimization algorithms tend to find a globally optimal solution, despite the highly non-convex landscape of the training loss function. Moreover, despite the extremely large amount of parameters, deep neural networks rarely over-fit, and can often generalize well to unseen data and achieve good test accuracy. Understanding these mysterious phenomena on the optimization and generalization of deep neural networks is one of the most fundamental problems in deep learning theory.

Recent breakthroughs have shed light on the optimization and generalization of deep neural networks (DNNs) under the over-parameterized setting, where the hidden layer width is extremely large (much larger than the number of training examples). It has been shown that with the standard random initialization, the training of over-parameterized deep neural networks can be characterized by a kernel function called neural tangent kernel (NTK) (Jacot et al., 2018; Arora et al., 2019b). In the neural tangent kernel regime (or lazy training regime (Chizat et al., 2019)), the neural network function behaves similarly as its first-order Taylor expansion at initialization (Jacot et al., 2018; Lee et al., 2019; Arora et al., 2019b; Cao and Gu, 2019), which enables feasible optimization and generalization analysis. In terms of optimization, a line of work (Du et al., 2019b; Allen-Zhu et al., 2019b; Zou et al., 2019; Zou and Gu, 2019) proved that for sufficiently wide neural networks, (stochastic) gradient descent (GD/SGD) can successfully find a global optimum of the training loss function. For generalization, Allen-Zhu et al. (2019a); Arora et al. (2019a); Cao and Gu (2019) established generalization bounds of neural networks trained with (stochastic) gradient descent, and showed that the neural networks can learn target functions in certain reproducing kernel Hilbert space (RKHS) or the corresponding random feature function class.

Although existing results in the neural tangent kernel regime have provided important insights into the learning of deep neural networks, they require the neural network to be extremely wide.

---

[*]Equal contribution.

The typical requirement on the network width is a high degree polynomial of the training sample size $n$ and the inverse of the target error $\epsilon^{-1}$. As there still remains a huge gap between such network width requirement and the practice, many attempts have been made to improve the over-parameterization condition under various conditions on the training data and model initialization (Oymak and Soltanolkotabi, 2019; Zou and Gu, 2019; Kawaguchi and Huang, 2019; Bai and Lee, 2019). For two-layer ReLU networks, a recent work (Ji and Telgarsky, 2020) showed that when the training data are well separated, polylogarithmic width is sufficient to guarantee good optimization and generalization performances. However, their results cannot be extended to deep ReLU networks since their proof technique largely relies on the fact that the network model is 1-homogeneous, which cannot be satisfied by DNNs. Therefore, whether deep neural networks can be learned with such a mild over-parameterization is still an open problem.

In this paper, we resolve this open problem by showing that polylogarithmic network width is sufficient to learn DNNs. In particular, unlike the existing works that require the DNNs to behave very close to a linear model (up to some small approximation error), we show that a constant linear approximation error is sufficient to establish nice optimization and generalization guarantees for DNNs. Thanks to the relaxed requirement on the linear approximation error, a milder condition on the network width and tighter bounds on the convergence rate and generalization error can be proved. We summarize our contributions as follows:

- We establish the global convergence guarantee of GD for training deep ReLU networks based on the so-called NTRF function class (Cao and Gu, 2019), a set of linear functions over random features. Specifically, we prove that GD can learn deep ReLU networks with width $m = \text{poly}(R)$ to compete with the best function in NTRF function class, where $R$ is the radius of the NTRF function class.

- We also establish the generalization guarantees for both GD and SGD in the same setting. Specifically, we prove a diminishing statistical error for a wide range of network width $m \in (\widetilde{\Omega}(1), \infty)$, while most of the previous generalization bounds in the NTK regime only works in the setting where the network width $m$ is much greater than the sample size $n$. Moreover, we establish $\widetilde{\mathcal{O}}(\epsilon^{-2})$ $\widetilde{\mathcal{O}}(\epsilon^{-1})$ sample complexities for GD and SGD respectively, which are tighter than existing bounds for learning deep ReLU networks (Cao and Gu, 2019), and match the best results when reduced to the two-layer cases (Arora et al., 2019b; Ji and Telgarsky, 2020).

- We further generalize our theoretical analysis to the scenarios with different data separability assumptions in the literature. We show if a large fraction of the training data are well separated, the best function in the NTRF function class with radius $R = \widetilde{\mathcal{O}}(1)$ can learn the training data with error up to $\epsilon$. This together with our optimization and generalization guarantees immediately suggests that deep ReLU networks can be learned with network width $m = \widetilde{\Omega}(1)$, which has a logarithmic dependence on the target error $\epsilon$ and sample size $n$. Compared with existing results (Cao and Gu, 2020; Ji and Telgarsky, 2020) which require all training data points to be separated in the NTK regime, our result is stronger since it allows the NTRF function class to misclassify a small proportion of the training data.

For the ease of comparison, we summarize our results along with the most related previous results in Table 1, in terms of data assumption, the over-parameterization condition and sample complexity. It can be seen that under data separation assumption (See Sections 4.1, 4.2), our result improves existing results for learning deep neural networks by only requiring a polylog($n, \epsilon^{-1}$) network width.

**Notation.** For two scalars $a$ and $b$, we denote $a \wedge b = \min\{a, b\}$. For a vector $\mathbf{x} \in \mathbb{R}^d$ we use $\|\mathbf{x}\|_2$ to denote its Euclidean norm. For a matrix $\mathbf{X}$, we use $\|\mathbf{X}\|_2$ and $\|\mathbf{X}\|_F$ to denote its spectral norm and Frobenius norm respectively, and denote by $\mathbf{X}_{ij}$ the entry of $\mathbf{X}$ at the $i$-th row and $j$-th column. Given two matrices $\mathbf{X}$ and $\mathbf{Y}$ with the same dimension, we denote $\langle \mathbf{X}, \mathbf{Y} \rangle = \sum_{i,j} \mathbf{X}_{ij} \mathbf{Y}_{ij}$.

Given a collection of matrices $\mathbf{W} = \{\mathbf{W}_1, \cdots, \mathbf{W}_L\} \in \otimes_{l=1}^{L} \mathbb{R}^{m_l \times m'_l}$ and a function $f(\mathbf{W})$ over $\otimes_{l=1}^{L} \mathbb{R}^{m_l \times m'_l}$, we define by $\nabla_{\mathbf{W}_l} f(\mathbf{W})$ the partial gradient of $f(\mathbf{W})$ with respect to $\mathbf{W}_l$ and denote $\nabla_{\mathbf{W}} f(\mathbf{W}) = \{\nabla_{\mathbf{W}_l} f(\mathbf{W})\}_{l=1}^{L}$. We also denote $\mathcal{B}(\mathbf{W}, \tau) = \{\mathbf{W}' : \max_{l \in [L]} \|\mathbf{W}'_l - \mathbf{W}_l\|_F \leqslant \tau\}$ for $\tau \geqslant 0$. For two collection of matrices $\mathbf{A} = \{\mathbf{A}_1, \cdots, \mathbf{A}_n\}$, $\mathbf{B} = \{\mathbf{B}_1, \cdots, \mathbf{B}_n\}$, we denote $\langle \mathbf{A}, \mathbf{B} \rangle = \sum_{i=1}^{n} \langle \mathbf{A}_i, \mathbf{B}_i \rangle$ and $\|\mathbf{A}\|_F^2 = \sum_{i=1}^{n} \|\mathbf{A}_i\|_F^2$.

Table 1: Comparison of neural network learning results in terms of over-parameterization condition and sample complexity. Here $\epsilon$ is the target error rate, $n$ is the sample size, $L$ is the network depth.

| | Assumptions | Algorithm | Over-para. Condition | Sample Complexity | Network |
|---|---|---|---|---|---|
| Zou et al. (2019) | Data nondegeneration | GD | $\tilde{\Omega}(n^{12}L^{16}(n^2 + \epsilon^{-1}))$ | - | Deep |
| **This paper** | Data nondegeneration | GD | $\tilde{\Omega}(L^{22}n^{12})$ | - | Deep |
| Cao and Gu (2020) | Data separation | GD | $\tilde{\Omega}(\epsilon^{-14}) \cdot e^{\Omega(L)}$ | $\tilde{\mathcal{O}}(\epsilon^{-4}) \cdot e^{O(L)}$ | Deep |
| Ji and Telgarsky (2020) | Data separation | GD | $\text{polylog}(n, \epsilon^{-1})$ | $\tilde{\mathcal{O}}(\epsilon^{-2})$ | Shallow |
| **This paper** | Data separation | GD | $\text{polylog}(n, \epsilon^{-1}) \cdot \text{poly}(L)$ | $\tilde{\mathcal{O}}(\epsilon^{-2}) \cdot e^{O(L)}$ | Deep |
| Cao and Gu (2019) | Data separation | SGD | $\tilde{\Omega}(\epsilon^{-14}) \cdot \text{poly}(L)$ | $\tilde{\mathcal{O}}(\epsilon^{-2}) \cdot \text{poly}(L)$ | Deep |
| Ji and Telgarsky (2020) | Data separation | SGD | $\text{polylog}(\epsilon^{-1})$ | $\tilde{\mathcal{O}}(\epsilon^{-1})$ | Shallow |
| **This paper** | Data separation | SGD | $\text{polylog}(\epsilon^{-1}) \cdot \text{poly}(L)$ | $\tilde{\mathcal{O}}(\epsilon^{-1}) \cdot \text{poly}(L)$ | Deep |

---

**Algorithm 1** Gradient descent with random initialization

**Input:** Number of iterations $T$, step size $\eta$, training set $S = \{(\mathbf{x}_i, y_i)_{i=1}^n\}$, initialization $\mathbf{W}^{(0)}$
**for** $t = 1, 2, \ldots, T$ **do**
    Update $\mathbf{W}^{(t)} = \mathbf{W}^{(t-1)} - \eta \cdot \nabla_{\mathbf{W}} L_S(\mathbf{W}^{(t-1)})$.
**end for**
**Output:** $\mathbf{W}^{(0)}, \ldots, \mathbf{W}^{(T)}$.

---

Given two sequences $\{x_n\}$ and $\{y_n\}$, we denote $x_n = \mathcal{O}(y_n)$ if $|x_n| \leqslant C_1|y_n|$ for some absolute positive constant $C_1$, $x_n = \Omega(y_n)$ if $|x_n| \geqslant C_2|y_n|$ for some absolute positive constant $C_2$, and $x_n = \Theta(y_n)$ if $C_3|y_n| \leqslant |x_n| \leqslant C_4|y_n|$ for some absolute constants $C_3, C_4 > 0$. We also use $\tilde{\mathcal{O}}(\cdot)$, $\tilde{\Omega}(\cdot)$ to hide logarithmic factors in $\mathcal{O}(\cdot)$ and $\Omega(\cdot)$ respectively. Additionally, we denote $x_n = \text{poly}(y_n)$ if $x_n = \mathcal{O}(y_n^D)$ for some positive constant $D$, and $x_n = \text{polylog}(y_n)$ if $x_n = \text{poly}(\log(y_n))$.

## 2 PRELIMINARIES ON LEARNING NEURAL NETWORKS

In this section, we introduce the problem setting in this paper, including definitions of the neural network and loss functions, and the training algorithms, i.e., GD and SGD with random initialization.

**Neural network function.** Given an input $\mathbf{x} \in \mathbb{R}^d$, the output of deep fully-connected ReLU network is defined as follows,

$$f_{\mathbf{W}}(\mathbf{x}) = m^{1/2}\mathbf{W}_L\sigma(\mathbf{W}_{L-1}\cdots\sigma(\mathbf{W}_1\mathbf{x})\cdots),$$

where $\mathbf{W}_1 \in \mathbb{R}^{m \times d}$, $\mathbf{W}_2, \cdots, \mathbf{W}_{L-1} \in \mathbb{R}^{m \times m}$, $\mathbf{W}_L \in \mathbb{R}^{1 \times m}$, and $\sigma(x) = \max\{0, x\}$ is the ReLU activation function. Here, without loss of generality, we assume the width of each layer is equal to $m$. Yet our theoretical results can be easily generalized to the setting with unequal width layers, as long as the smallest width satisfies our overparameterization condition. We denote the collection of all weight matrices as $\mathbf{W} = \{\mathbf{W}_1, \ldots, \mathbf{W}_L\}$.

**Loss function.** Given training dataset $\{\mathbf{x}_i, y_i\}_{i=1,\ldots,n}$ with input $\mathbf{x}_i \in \mathbb{R}^d$ and output $y_i \in \{-1, +1\}$, we define the training loss function as

$$L_S(\mathbf{W}) = \frac{1}{n}\sum_{i=1}^n L_i(\mathbf{W}),$$

where $L_i(\mathbf{W}) = \ell(y_i f_{\mathbf{W}}(\mathbf{x}_i)) = \log(1 + \exp(-y_i f_{\mathbf{W}}(\mathbf{x}_i)))$ is defined as the cross-entropy loss.

**Algorithms.** We consider both GD and SGD with Gaussian random initialization. These two algorithms are displayed in Algorithms 1 and 2 respectively. Specifically, the entries in $\mathbf{W}_1^{(0)}, \cdots, \mathbf{W}_{L-1}^{(0)}$ are generated independently from univariate Gaussian distribution $N(0, 2/m)$ and the entries in $\mathbf{W}_L^{(0)}$ are generated independently from $N(0, 1/m)$. For GD, we consider using the full gradient to update the model parameters. For SGD, we use a new training data point in each iteration.

Note that our initialization method in Algorithms 1, 2 is the same as the widely used He initialization (He et al., 2015). Our neural network parameterization is also consistent with the parameterization used in prior work on NTK (Jacot et al., 2018; Allen-Zhu et al., 2019b; Du et al., 2019a; Arora et al., 2019b; Cao and Gu, 2019).

---

**Algorithm 2** Stochastic gradient desecent (SGD) with random initialization

---

**Input:** Number of iterations $n$, step size $\eta$, initialization $\mathbf{W}^{(0)}$
**for** $i = 1, 2, \ldots, n$ **do**
    Draw $(\mathbf{x}_i, y_i)$ from $\mathcal{D}$ and compute the corresponding gradient $\nabla_{\mathbf{W}} L_i(\mathbf{W}^{(i-1)})$.
    Update $\mathbf{W}^{(i)} = \mathbf{W}^{(i-1)} - \eta \cdot \nabla_{\mathbf{W}} L_i(\mathbf{W}^{(i-1)})$.
**end for**
**Output:** Randomly choose $\widehat{\mathbf{W}}$ uniformly from $\{\mathbf{W}^{(0)}, \ldots, \mathbf{W}^{(n-1)}\}$.

---

## 3 MAIN THEORY

In this section, we present the optimization and generalization guarantees of GD and SGD for learning deep ReLU networks. We first make the following assumption on the training data points.

**Assumption 3.1.** All training data points satisfy $\|\mathbf{x}_i\|_2 = 1$, $i = 1, \ldots, n$.

This assumption has been widely made in many previous works (Allen-Zhu et al., 2019b;c; Du et al., 2019b;a; Zou et al., 2019) in order to simplify the theoretical analysis. This assumption can be relaxed to be upper bounded and lower bounded by some constant.

In the following, we give the definition of Neural Tangent Random Feature (NTRF) (Cao and Gu, 2019), which characterizes the functions learnable by over-parameterized ReLU networks.

**Definition 3.2** (Neural Tangent Random Feature, (Cao and Gu, 2019)). Let $\mathbf{W}^{(0)}$ be the initialization weights, and $F_{\mathbf{W}^{(0)}, \mathbf{W}}(\mathbf{x}) = f_{\mathbf{W}^{(0)}}(\mathbf{x}) + \langle \nabla f_{\mathbf{W}^{(0)}}(\mathbf{x}), \mathbf{W} - \mathbf{W}^{(0)} \rangle$ be a function with respect to the input $\mathbf{x}$. Then the NTRF function class is defined as follows

$$\mathcal{F}(\mathbf{W}^{(0)}, R) = \big\{ F_{\mathbf{W}^{(0)}, \mathbf{W}}(\cdot) : \mathbf{W} \in \mathcal{B}(\mathbf{W}^{(0)}, R \cdot m^{-1/2}) \big\}.$$

The function class $F_{\mathbf{W}^{(0)}, \mathbf{W}}(\mathbf{x})$ consists of linear models over random features defined based on the network gradients at the initialization. Therefore it captures the key "almost linear" property of wide neural networks in the NTK regime (Lee et al., 2019; Cao and Gu, 2019). In this paper, we use the NTRF function class as a reference class to measure the difficulty of a learning problem. In what follows, we deliver our main theoretical results regarding the optimization and generalization guarantees of learning deep ReLU networks. We study both GD and SGD with random initialization (presented in Algorithms 1 and 2).

### 3.1 GRADIENT DESCENT

The following theorem establishes the optimization guarantee of GD for training deep ReLU networks for binary classification.

**Theorem 3.3.** For $\delta, R > 0$, let $\epsilon_{\text{NTRF}} = \inf_{F \in \mathcal{F}(\mathbf{W}^{(0)}, R)} n^{-1} \sum_{i=1}^n \ell[y_i F(\mathbf{x}_i)]$ be the minimum training loss achievable by functions in $\mathcal{F}(\mathbf{W}^{(0)}, R)$. Then there exists

$$m^*(\delta, R, L) = \widetilde{\mathcal{O}}\big(\text{poly}(R, L) \cdot \log^{4/3}(n/\delta)\big),$$

such that if $m \geqslant m^*(\delta, R, L)$, with probability at least $1 - \delta$ over the initialization, GD with step size $\eta = \Theta(L^{-1} m^{-1})$ can train a neural network to achieve at most $3\epsilon_{\text{NTRF}}$ training loss within $T = \mathcal{O}\big(L^2 R^2 \epsilon_{\text{NTRF}}^{-1}\big)$ iterations.

Theorem 3.3 shows that the deep ReLU network trained by GD can compete with the best function in the NTRF function class $\mathcal{F}(\mathbf{W}^{(0)}, R)$ if the network width has a polynomial dependency in $R$ and $L$ and a logarithmic dependency in $n$ and $1/\delta$. Moreover, if the NTRF function class with $R = \widetilde{\mathcal{O}}(1)$ can learn the training data well (i.e., $\epsilon_{\text{NTRF}}$ is less than a small target error $\epsilon$), a polylogarithmic (in terms of $n$ and $\epsilon^{-1}$) network width suffices to guarantee the global convergence of GD, which directly improves over-paramterization condition in the most related work (Cao and Gu, 2019). Besides, we remark here that this assumption on the NTRF function class can be easily satisfied when the training data admits certain separability conditions, which we discuss in detail in Section 4.

Compared with the results in (Ji and Telgarsky, 2020) which give similar network width requirements for two-layer networks, our result works for deep networks. Moreover, while Ji and Telgarsky (2020)

essentially required all training data to be separable by a function in the NTRF function class with a constant margin, our result does not require such data separation assumptions, and allows the NTRF function class to misclassify a small proportion of the training data points[*].

We now characterize the generalization performance of neural networks trained by GD. We denote $L_{\mathcal{D}}^{0-1}(\mathbf{W}) = \mathbb{E}_{(\mathbf{x},y)\sim\mathcal{D}}[\mathbb{1}\{f_{\mathbf{W}}(\mathbf{x})\cdot y < 0\}]$ as the expected 0-1 loss (i.e., expected error) of $f_{\mathbf{W}}(\mathbf{x})$.

**Theorem 3.4.** Under the same assumptions as Theorem 3.3, with probability at least $1-\delta$, the iterate $\mathbf{W}^{(t)}$ of Algorithm 1 satisfies that

$$L_{\mathcal{D}}^{0-1}(\mathbf{W}^{(t)}) \leqslant 2L_S(\mathbf{W}^{(t)}) + \tilde{\mathcal{O}}\left(4^L L^2 R\sqrt{\frac{m}{n}} \wedge \left(\frac{L^{3/2}R}{\sqrt{n}} + \frac{L^{11/3}R^{4/3}}{m^{1/6}}\right)\right) + \mathcal{O}\left(\sqrt{\frac{\log(1/\delta)}{n}}\right)$$

for all $t = 0, \ldots, T$.

Theorem 3.4 shows that the test error of the trained neural network can be bounded by its training error plus statistical error terms. Note that the statistical error terms is in the form of a minimum between two terms $4^L L^2 R\sqrt{m/n}$ and $L^{3/2}R/\sqrt{n} + L^{11/3}R^{4/3}/m^{1/6}$. Depending on the network width $m$, one of these two terms will be the dominating term and diminishes for large $n$: (1) if $m = o(n)$, the statistical error will be $4^L L^2 R\sqrt{m/n}$, and diminishes as $n$ increases; and (2) if $m = \Omega(n)$, the statistical error is $L^{3/2}R/\sqrt{n} + L^{11/3}R^{4/3}/m^{1/6}$, and again goes to zero as $n$ increases. Moreover, in this paper we have a specific focus on the setting $m = \tilde{\mathcal{O}}(1)$, under which Theorem 3.4 gives a statistical error of order $\tilde{\mathcal{O}}(n^{-1/2})$. This distinguishes our result from previous generalization bounds for deep networks (Cao and Gu, 2020; 2019), which cannot be applied to the setting $m = \tilde{\mathcal{O}}(1)$.

We note that for two-layer ReLU networks (i.e., $L = 2$) Ji and Telgarsky (2020) proves a tighter $\tilde{O}(1/n^{1/2})$ generalization error bound regardless of the neural networks width $m$, while our result (Theorem 3.4), in the two-layer case, can only give $\tilde{O}(1/n^{1/2})$ generalization error bound when $m = \tilde{O}(1)$ or $m = \tilde{\Omega}(n^3)$. However, different from our proof technique that basically uses the (approximated) linearity of the neural network function, their proof technique largely relies on the 1-homogeneous property of the neural network, which restricted their theory in two-layer cases. An interesting research direction is to explore whether a $\tilde{O}(1/n^{1/2})$ generalization error bound can be also established for deep networks (regardless of the network width), which we will leave it as a future work.

## 3.2 STOCHASTIC GRADIENT DESCENT

Here we study the performance of SGD for training deep ReLU networks. The following theorem establishes a generalization error bound for the output of SGD.

**Theorem 3.5.** For $\delta, R > 0$, let $\epsilon_{\text{NTRF}} = \inf_{F \in \mathcal{F}(\mathbf{W}^{(0)}, R)} n^{-1} \sum_{i=1}^{n} \ell[y_i F(\mathbf{x}_i)]$ be the minimum training loss achievable by functions in $\mathcal{F}(\mathbf{W}^{(0)}, R)$. Then there exists

$$m^*(\delta, R, L) = \tilde{\mathcal{O}}\big(\text{poly}(R, L) \cdot \log^{4/3}(n/\delta)\big),$$

such that if $m \geqslant m^*(\delta, R, L)$, with probability at least $1-\delta$, SGD with step size $\eta = \Theta\big(m^{-1} \cdot (LR^2 n^{-1}\epsilon_{\text{NTRF}}^{-1} \wedge L^{-1})\big)$ achieves

$$\mathbb{E}[L_{\mathcal{D}}^{0-1}(\widehat{\mathbf{W}})] \leqslant \frac{8L^2 R^2}{n} + \frac{8\log(2/\delta)}{n} + 24\epsilon_{\text{NTRF}},$$

where the expectation is taken over the uniform draw of $\widehat{\mathbf{W}}$ from $\{\mathbf{W}^{(0)}, \ldots, \mathbf{W}^{(n-1)}\}$.

For any $\epsilon > 0$, Theorem 3.5 gives a $\tilde{\mathcal{O}}(\epsilon^{-1})$ sample complexity for deep ReLU networks trained with SGD to achieve $O(\epsilon_{\text{NTRF}} + \epsilon)$ test error. Our result extends the result for two-layer networks proved in (Ji and Telgarsky, 2020) to multi-layer networks. Theorem 3.5 also provides sharper results compared with Allen-Zhu et al. (2019a); Cao and Gu (2019) in two aspects: (1) the sample complexity is improved from $n = \tilde{\mathcal{O}}(\epsilon^{-2})$ to $n = \tilde{\mathcal{O}}(\epsilon^{-1})$; and (2) the overparamterization condition is improved from $m \geqslant \text{poly}(\epsilon^{-1})$ to $m \geqslant \tilde{\Omega}(1)$.

---

[*]A detailed discussion is given in Section 4.2.

## 4   DISCUSSION ON THE NTRF CLASS

Our theoretical results in Section 3 rely on the radius (i.e., $R$) of the NTRF function class $\mathcal{F}(\mathbf{W}^{(0)}, R)$ and the minimum training loss achievable by functions in $\mathcal{F}(\mathbf{W}^{(0)}, R)$, i.e., $\epsilon_{\text{NTRF}}$. Note that a larger $R$ naturally implies a smaller $\epsilon_{\text{NTRF}}$, but also leads to worse conditions on $m$. In this section, for any (arbitrarily small) target error rate $\epsilon > 0$, we discuss various data assumptions studied in the literature under which our results can lead to $\mathcal{O}(\epsilon)$ training/test errors, and specify the network width requirement.

### 4.1   DATA SEPARABILITY BY NEURAL TANGENT RANDOM FEATURE

In this subsection, we consider the setting where a large fraction of the training data can be linearly separated by the neural tangent random features. The assumption is stated as follows.

**Assumption 4.1.** There exists a collection of matrices $\mathbf{U}^* = \{\mathbf{U}_1^*, \cdots, \mathbf{U}_L^*\}$ satisfying $\sum_{l=1}^{L} \|\mathbf{U}_l^*\|_F^2 = 1$, such that for at least $(1 - \rho)$ fraction of training data we have

$$y_i \langle \nabla f_{\mathbf{W}^{(0)}}(\mathbf{x}_i), \mathbf{U}^* \rangle \geqslant m^{1/2}\gamma,$$

where $\gamma$ is an absolute positive constant[†] and $\rho \in [0, 1)$.

The following corollary provides an upper bound of $\epsilon_{\text{NTRF}}$ under Assumption 4.1 for some $R$.

**Proposition 4.2.** Under Assumption 4.1, for any $\epsilon, \delta > 0$, if $R \geqslant C\big[\log^{1/2}(n/\delta) + \log(1/\epsilon)\big]/\gamma$ for some absolute constant $C$, then with probability at least $1 - \delta$,

$$\epsilon_{\text{NTRF}} := \inf_{F \in \mathcal{F}(\mathbf{W}^{(0)}, R)} n^{-1} \sum_{i=1}^{n} \ell\big(y_i F(\mathbf{x}_i)\big) \leqslant \epsilon + \rho \cdot \mathcal{O}(R).$$

Proposition 4.2 covers the setting where the NTRF function class is allowed to misclassify training data, while most of existing work typically assumes that all training data can be perfectly separated with constant margin (i.e., $\rho = 0$) (Ji and Telgarsky, 2020; Shamir, 2020). Our results show that for sufficiently small misclassification ratio $\rho = \mathcal{O}(\epsilon)$, we have $\epsilon_{\text{NTRF}} = \widetilde{O}(\epsilon)$ by choosing the radius parameter $R$ logarithimic in $n$, $\delta^{-1}$, and $\epsilon^{-1}$. Substituting this result into Theorems 3.3, 3.4 and 3.5, it can be shown that a neural network with width $m = \text{poly}(L, \log(n/\delta), \log(1/\epsilon)))$ suffices to guarantee good optimization and generalization performances for both GD and SGD. Consequently, we can obtain that the bounds on the test error for GD and SGD are $\widetilde{\mathcal{O}}(n^{-1/2})$ and $\widetilde{\mathcal{O}}(n^{-1})$ respectively.

### 4.2   DATA SEPARABILITY BY SHALLOW NEURAL TANGENT MODEL

In this subsection, we study the data separation assumption made in Ji and Telgarsky (2020) and show that our results cover this particular setting. We first restate the assumption as follows.

**Assumption 4.3.** There exists $\overline{\mathbf{u}}(\cdot) : \mathbb{R}^d \to \mathbb{R}^d$ and $\gamma \geqslant 0$ such that $\|\overline{\mathbf{u}}(\mathbf{z})\|_2 \leqslant 1$ for all $\mathbf{z} \in \mathbb{R}^d$, and

$$y_i \int_{\mathbb{R}^d} \sigma'(\langle \mathbf{z}, \mathbf{x}_i \rangle) \cdot \langle \overline{\mathbf{u}}(\mathbf{z}), \mathbf{x}_i \rangle \mathrm{d}\mu_N(\mathbf{z}) \geqslant \gamma$$

for all $i \in [n]$, where $\mu_N(\cdot)$ denotes the standard normal distribution.

Assumption 4.3 is related to the linear separability of the gradients of the first layer parameters at random initialization, where the randomness is replaced with an integral by taking the infinite width limit. Note that similar assumptions have also been studied in (Cao and Gu, 2020; Nitanda and Suzuki, 2019; Frei et al., 2019). The assumption made in (Cao and Gu, 2020; Frei et al., 2019) uses gradients with respect to the second layer weights instead of the first layer ones. In the following, we mainly focus on Assumption 4.3, while our result can also be generalized to cover the setting in (Cao and Gu, 2019; Frei et al., 2019).

---

[†]The factor $m^{1/2}$ is introduced here since $\|\nabla_{\mathbf{W}^{(0)}} f(\mathbf{x}_i)\|_F$ is typically of order $O(m^{1/2})$.

In order to make a fair comparison, we reduce our results for multilayer networks to the two-layer setting. In this case, the neural network function takes form

$$f_{\mathbf{W}}(\mathbf{x}) = m^{1/2}\mathbf{W}_2\sigma(\mathbf{W}_1\mathbf{x}).$$

Then we provide the following proposition, which states that Assumption 4.3 implies a certain choice of $R = \widetilde{\mathcal{O}}(1)$ such the the minimum training loss achieved by the function in the NTRF function class $\mathcal{F}(\mathbf{W}^{(0)}, R)$ satisfies $\epsilon_{\text{NTRF}} = O(\epsilon)$, where $\epsilon$ is the target error.

**Proposition 4.4.** Suppose the training data satisfies Assumption 4.3. For any $\epsilon, \delta > 0$, let $R = C\big[\log(n/\delta) + \log(1/\epsilon)\big]/\gamma$ for some large enough absolute constant $C$. If the neural network width satisfies $m = \Omega\big(\log(n/\delta)/\gamma^2\big)$, then with probability at least $1 - \delta$, there exist $F_{\mathbf{W}^{(0)}, \overline{\mathbf{W}}}(\mathbf{x}_i) \in \mathcal{F}(\mathbf{W}^{(0)}, R)$ such that $\ell\big(y_i \cdot F_{\mathbf{W}^{(0)}, \overline{\mathbf{W}}}(\mathbf{x}_i)\big) \leqslant \epsilon, \forall i \in [n]$.

Proposition 4.4 shows that under Assumption 4.3, there exists $F_{\mathbf{W}^{(0)}, \overline{\mathbf{W}}}(\cdot) \in \mathcal{F}(\mathbf{W}^{(0)}, R)$ with $R = \widetilde{\mathcal{O}}(1/\gamma)$ such that the cross-entropy loss of $F_{\mathbf{W}^{(0)}, \overline{\mathbf{W}}}(\cdot)$ at each training data point is bounded by $\epsilon$. This implies that $\epsilon_{\text{NTRF}} \leqslant \epsilon$. Moreover, by applying Theorem 3.3 with $L = 2$, the condition on the neural network width becomes $m = \widetilde{\Omega}(1/\gamma^8)^{\ddagger}$, which matches the results proved in Ji and Telgarsky (2020). Moreover, plugging these results on $m$ and $\epsilon_{\text{NTRF}}$ into Theorems 3.4 and 3.5, we can conclude that the bounds on the test error for GD and SGD are $\widetilde{\mathcal{O}}(n^{-1/2})$ and $\widetilde{\mathcal{O}}(n^{-1})$ respectively.

### 4.3 CLASS-DEPENDENT DATA NONDEGENERATION

In previous subsections, we have shown that under certain data separation conditions $\epsilon_{\text{NTRF}}$ can be sufficiently small while the corresponding NTRF function class has $R$ of order $\widetilde{\mathcal{O}}(1)$. Thus neural networks with polylogarithmic width enjoy nice optimization and generalization guarantees. In this part, we consider the following much milder data separability assumption made in Zou et al. (2019).

**Assumption 4.5.** For all $i \neq i'$ if $y_i \neq y_{i'}$, then $\|\mathbf{x}_i - \mathbf{x}_j\|_2 \geqslant \phi$ for some absolute constant $\phi$.

In contrast to the conventional data nondegeneration assumption (i.e., no duplicate data points) made in Allen-Zhu et al. (2019b); Du et al. (2019b;a); Zou and Gu (2019)$^{\S}$, Assumption 4.5 only requires that the data points from different classes are nondegenerate, thus we call it class-dependent data nondegeneration.

We have the following proposition which shows that Assumption 4.5 also implies the existence of a good function that achieves $\epsilon$ training error, in the NTRF function class with a certain choice of $R$.

**Proposition 4.6.** Under Assumption 4.5, if

$$R = \Omega\big(n^{3/2}\phi^{-1/2}\log(n\delta^{-1}\epsilon^{-1})\big), \qquad m = \widetilde{\Omega}\big(L^{22}n^{12}\phi^{-4}\big),$$

we have $\epsilon_{\text{NTRF}} \leqslant \epsilon$ with probability at least $1 - \delta$.

Proposition 4.6 suggests that under Assumption 4.5, in order to guarantee $\epsilon_{\text{NTRF}} \leqslant \epsilon$, the size of NTRF function class needs to be $\Omega(n^{3/2})$. Plugging this into Theorems 3.4 and 3.5 leads to vacuous bounds on the test error. This makes sense since Assumption 4.5 basically covers the "random label" setting, which is impossible to be learned with small generalization error. Moreover, we would like to point out our theoretical analysis leads to a sharper over-parameterization condition than that proved in Zou et al. (2019), i.e., $m = \widetilde{\Omega}\big(n^{14}L^{16}\phi^{-4} + n^{12}L^{16}\phi^{-4}\epsilon^{-1}\big)$, if the network depth satisfies $L \leqslant \widetilde{\mathcal{O}}(n^{1/3} \vee \epsilon^{-1/6})$.

## 5 PROOF SKETCH OF THE MAIN THEORY

In this section, we introduce a key technical lemma in Section 5.1, based on which we provide a proof sketch of Theorems 3.3. The full proof of all our results can be found in the appendix.

---

$^{\ddagger}$We have shown in the proof of Theorem 3.3 that $m = \widetilde{\Omega}(R^8)$ (see (A.1) for more detail).

$^{\S}$Specifically, Allen-Zhu et al. (2019b); Zou and Gu (2019) require that any two data points (rather than data points from different classes) are separated by a positive distance. Zou and Gu (2019) shows that this assumption is equivalent to those made in Du et al. (2019b;a), which require that the composite kernel matrix is strictly positive definite.

## 5.1 A KEY TECHNICAL LEMMA

Here we introduce a key technical lemma used in the proof of Theorem 3.3.

Our proof is based on the key observation that near initialization, the neural network function can be approximated by its first-order Taylor expansion. In the following, we first give the definition of the linear approximation error in a $\tau$-neighborhood around initialization.

$$\epsilon_{\text{app}}(\tau) := \sup_{i=1,\ldots,n} \sup_{\mathbf{W}',\mathbf{W} \in \mathcal{B}(\mathbf{W}^{(0)},\tau)} \left| f_{\mathbf{W}'}(\mathbf{x}_i) - f_{\mathbf{W}}(\mathbf{x}_i) - \langle \nabla f_{\mathbf{W}}(\mathbf{x}_i), \mathbf{W}' - \mathbf{W} \rangle \right|.$$

If all the iterates of GD stay inside a neighborhood around initialization with small linear approximation error, then we may expect that the training of neural networks should be similar to the training of the corresponding linear model, where standard optimization techniques can be applied. Motivated by this, we also give the following definition on the gradient upper bound of neural networks around initialization, which is related to the Lipschitz constant of the optimization objective function.

$$M(\tau) := \sup_{i=1,\ldots,n} \sup_{l=1,\ldots,L} \sup_{\mathbf{W} \in \mathcal{B}(\mathbf{W}^{(0)},\tau)} \|\nabla_{\mathbf{W}_l} f_{\mathbf{W}}(\mathbf{x}_i)\|_F.$$

By definition, we can choose $\mathbf{W}^* \in \mathcal{B}(\mathbf{W}^{(0)}, Rm^{-1/2})$ such that $n^{-1} \sum_{i=1}^{n} \ell(y_i F_{\mathbf{W}^{(0)},\mathbf{W}^*}(\mathbf{x}_i)) = \epsilon_{\text{NTRF}}$. Then we have the following lemma.

**Lemma 5.1.** Set $\eta = \mathcal{O}(L^{-1}M(\tau)^{-2})$. Suppose that $\mathbf{W}^* \in \mathcal{B}(\mathbf{W}^{(0)}, \tau)$ and $\mathbf{W}^{(t)} \in \mathcal{B}(\mathbf{W}^{(0)}, \tau)$ for all $0 \leqslant t \leqslant t' - 1$. Then it holds that

$$\frac{1}{t'} \sum_{t=0}^{t'-1} L_{\mathcal{S}}(\mathbf{W}^{(t)}) \leqslant \frac{\|\mathbf{W}^{(0)} - \mathbf{W}^*\|_F^2 - \|\mathbf{W}^{(t')} - \mathbf{W}^*\|_F^2 + 2t'\eta\epsilon_{\text{NTRF}}}{t'\eta(\frac{3}{2} - 4\epsilon_{\text{app}}(\tau))}.$$

Lemma 5.1 plays a central role in our proof. In specific, if $\mathbf{W}^{(t)} \in \mathcal{B}(\mathbf{W}^{(0)}, \tau)$ for all $t \leqslant t'$, then Lemma 5.1 implies that the average training loss is in the same order of $\epsilon_{\text{NTRF}}$ as long as the linear approximation error $\epsilon_{\text{app}}(\tau)$ is bounded by a positive constant. This is in contrast to the proof in Cao and Gu (2019), where $\epsilon_{\text{app}}(\tau)$ appears as an additive term in the upper bound of the training loss, thus requiring $\epsilon_{\text{app}}(\tau) = \mathcal{O}(\epsilon_{\text{NTRF}})$ to achieve the same error bound as in Lemma 5.1. Since we can show that $\epsilon_{\text{app}} = \widetilde{\mathcal{O}}(m^{-1/6})$ (See Section A.1), this suggests that $m = \widetilde{\Omega}(1)$ is sufficient to make the average training loss in the same order of $\epsilon_{\text{NTRF}}$.

Compared with the recent results for two-layer networks by Ji and Telgarsky (2020), Lemma 5.1 is proved with different techniques. In specific, the proof by Ji and Telgarsky (2020) relies on the 1-homogeneous property of the ReLU activation function, which limits their analysis to two-layer networks with fixed second layer weights. In comparison, our proof does not rely on homogeneity, and is purely based on the linear approximation property of neural networks and some specific properties of the loss function. Therefore, our proof technique can handle deep networks, and is potentially applicable to non-ReLU activation functions and other network architectures (e.g, Convolutional neural networks and Residual networks).

## 5.2 PROOF SKETCH OF THEOREM 3.3

Here we provide a proof sketch of Theorem 3.3. The proof consists of two steps: (i) showing that all $T$ iterates stay close to initialization, and (ii) bounding the empirical loss achieved by gradient descent. Both of these steps are proved based on Lemma 5.1.

*Proof sketch of Theorem 3.3.* Recall that we choose $\mathbf{W}^* \in \mathcal{B}(\mathbf{W}^{(0)}, Rm^{-1/2})$ such that $n^{-1} \sum_{i=1}^{n} \ell(y_i F_{\mathbf{W}^{(0)},\mathbf{W}^*}(\mathbf{x}_i)) = \epsilon_{\text{NTRF}}$. We set $\tau = \widetilde{\mathcal{O}}(L^{1/2}m^{-1/2}R)$, which is chosen slightly larger than $m^{-1/2}R$ since Lemma 5.1 requires the region $\mathcal{B}(\mathbf{W}^{(0)}, \tau)$ to include both $\mathbf{W}^*$ and $\{\mathbf{W}^{(t)}\}_{t=0,\ldots,t'}$. Then by Lemmas 4.1 and B.3 in Cao and Gu (2019) we know that $\epsilon_{\text{app}}(\tau) = \widetilde{\mathcal{O}}(\tau^{4/3}m^{1/2}L^3) = \widetilde{\mathcal{O}}(R^{4/3}L^{11/3}m^{-1/6})$. Therefore, we can set $m = \widetilde{\Omega}(R^8 L^{22})$ to ensure that $\epsilon_{\text{app}}(\tau) \leqslant 1/8$.

Then we proceed to show that all iterates stay inside the region $\mathcal{B}(\mathbf{W}^{(0)}, \tau)$. Since the L.H.S. of Lemma 5.1 is strictly positive and $\epsilon_{\text{app}}(\tau) \leqslant 1/8$, we have for all $t \leqslant T$,

$$\|\mathbf{W}^{(0)} - \mathbf{W}^*\|_F^2 - \|\mathbf{W}^{(t)} - \mathbf{W}^*\|_F^2 \geqslant -2t\eta\epsilon_{\text{NTRF}},$$

which gives an upper bound of $\|\mathbf{W}^{(t)} - \mathbf{W}^*\|_F$. Then by the choice of $\eta$, $T$, triangle inequality, and a simple induction argument, we see that $\|\mathbf{W}^{(t)} - \mathbf{W}^{(0)}\|_F \leqslant m^{-1/2}R + \sqrt{2T\eta\epsilon_{\mathrm{NTRF}}} = \widetilde{\mathcal{O}}(L^{1/2}m^{-1/2}R)$, which verifies that $\mathbf{W}^{(t)} \in \mathcal{B}(\mathbf{W}^{(0)}, \tau)$ for $t = 0, \ldots, T - 1$.

The second step is to show that GD can find a neural network with at most $3\epsilon_{\mathrm{NTRF}}$ training loss within $T$ iterations. To show this, by the bound given in Lemma 5.1 with $\epsilon_{\mathrm{app}} \leqslant 1/8$, we drop the terms $\|\mathbf{W}^{(t)} - \mathbf{W}^*\|_F^2$ and rearrange the inequality to obtain

$$\frac{1}{T} \sum_{t=0}^{T-1} L_S(\mathbf{W}^{(t)}) \leqslant \frac{1}{\eta T}\|\mathbf{W}^{(0)} - \mathbf{W}^*\|_F^2 + 2\epsilon_{\mathrm{NTRF}}.$$

We see that $T$ is large enough to ensure that the first term in the bound above is smaller than $\epsilon_{\mathrm{NTRF}}$. This implies that the best iterate among $\mathbf{W}^{(0)}, \ldots, \mathbf{W}^{(T-1)}$ achieves an empirical loss at most $3\epsilon_{\mathrm{NTRF}}$. □

## 6 CONCLUSION

In this paper, we established the global convergence and generalization error bounds of GD and SGD for training deep ReLU networks for the binary classification problem. We show that a network width condition that is polylogarithmic in the sample size $n$ and the inverse of target error $\epsilon^{-1}$ is sufficient to guarantee the learning of deep ReLU networks. Our results resolve an open question raised in Ji and Telgarsky (2020).

### ACKNOWLEDGEMENT

We would like to thank the anonymous reviewers for their helpful comments. ZC, YC and QG are partially supported by the National Science Foundation CAREER Award 1906169, IIS-2008981 and Salesforce Deep Learning Research Award. DZ is supported by the Bloomberg Data Science Ph.D. Fellowship. The views and conclusions contained in this paper are those of the authors and should not be interpreted as representing any funding agencies.

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

# A    PROOF OF MAIN THEOREMS

In this section we provide the full proof of Theorems 3.3, 3.4 and 3.5.

## A.1    PROOF OF THEOREM 3.3

We first provide the following lemma which is useful in the subsequent proof.

**Lemma A.1** (Lemmas 4.1 and B.3 in Cao and Gu (2019))**.** There exists an absolute constant $\kappa$ such that, with probability at least $1 - \mathcal{O}(nL^2)\exp[-\Omega(m\tau^{2/3}L)]$, for any $\tau \leqslant \kappa L^{-6}[\log(m)]^{-3/2}$, it holds that

$$\epsilon_{\mathrm{app}}(\tau) \leqslant \widetilde{\mathcal{O}}\big(\tau^{4/3}L^3 m^{1/2}\big), \quad M(\tau) \leqslant \widetilde{\mathcal{O}}(\sqrt{m}).$$

*Proof of Theorem 3.3.* Recall that $\mathbf{W}^*$ is chosen such that

$$\frac{1}{n}\sum_{i=1}^{n}\ell\big(y_i F_{\mathbf{W}^{(0)},\mathbf{W}^*}(\mathbf{x}_i)\big) = \epsilon_{\mathrm{NTRF}}$$

and $\mathbf{W}^* \in \mathcal{B}(\mathbf{W}^{(0)}, Rm^{-1/2})$. Note that to apply Lemma 5.1, we need the region $\mathcal{B}(\mathbf{W}^{(0)}, \tau)$ to include both $\mathbf{W}^*$ and $\{\mathbf{W}^{(t)}\}_{t=0,\ldots,t'}$. This motivates us to set $\tau = \widetilde{\mathcal{O}}(L^{1/2}m^{-1/2}R)$, which is slightly larger than $m^{-1/2}R$. With this choice of $\tau$, by Lemma A.1 we have $\epsilon_{\mathrm{app}}(\tau) = \widetilde{\mathcal{O}}(\tau^{4/3}m^{1/2}L^3) = \widetilde{\mathcal{O}}(R^{4/3}L^{11/3}m^{-1/6})$. Therefore, we can set

$$m = \widetilde{\Omega}(R^8 L^{22}) \tag{A.1}$$

to ensure that $\epsilon_{\mathrm{app}}(\tau) \leqslant 1/8$, where $\widetilde{\Omega}(\cdot)$ hides polylogarithmic dependencies on network depth $L$, NTRF function class size $R$, and failure probability parameter $\delta$. Then by Lemma 5.1, we have with probability at least $1 - \delta$, we have

$$\|\mathbf{W}^{(0)} - \mathbf{W}^*\|_F^2 - \|\mathbf{W}^{(t')} - \mathbf{W}^*\|_F^2 \geqslant \eta \sum_{t=0}^{t'-1} L_S(\mathbf{W}^{(t)}) - 2t'\eta\epsilon_{\mathrm{NTRF}} \tag{A.2}$$

as long as $\mathbf{W}^{(0)}, \ldots, \mathbf{W}^{(t'-1)} \in \mathcal{B}(\mathbf{W}^{(0)}, \tau)$. In the following proof we choose $\eta = \Theta(L^{-1}m^{-1})$ and $T = \lceil LR^2 m^{-1}\eta^{-1}\epsilon_{\mathrm{NTRF}}^{-1}\rceil$.

We prove the theorem by two steps: 1) we show that all iterates $\{\mathbf{W}^{(0)}, \cdots, \mathbf{W}^{(T)}\}$ will stay inside the region $\mathcal{B}(\mathbf{W}^{(0)}, \tau)$; and 2) we show that GD can find a neural network with at most $3\epsilon_{\mathrm{NTRF}}$ training loss within $T$ iterations.

**All iterates stay inside $\mathcal{B}(\mathbf{W}^{(0)}, \tau)$.** We prove this part by induction. Specifically, given $t' \leqslant T$, we assume the hypothesis $\mathbf{W}^{(t)} \in \mathcal{B}(\mathbf{W}^{(0)}, \tau)$ holds for all $t < t'$ and prove that $\mathbf{W}^{(t')} \in \mathcal{B}(\mathbf{W}^{(0)}, \tau)$. First, it is clear that $\mathbf{W}^{(0)} \in \mathcal{B}(\mathbf{W}^{(0)}, \tau)$. Then by (A.2) and the fact that $L_S(\mathbf{W}) \geqslant 0$, we have

$$\|\mathbf{W}^{(t')} - \mathbf{W}^*\|_F^2 \leqslant \|\mathbf{W}^{(0)} - \mathbf{W}^*\|_F^2 + 2\eta t'\epsilon_{\mathrm{NTRF}}$$

Note that $T = \lceil LR^2 m^{-1}\eta^{-1}\epsilon_{\mathrm{NTRF}}^{-1}\rceil$ and $\mathbf{W}^* \in \mathcal{B}(\mathbf{W}^{(0)}, R \cdot m^{-1/2})$, we have

$$\sum_{l=1}^{L}\|\mathbf{W}_l^{(t')} - \mathbf{W}_l^*\|_F^2 = \|\mathbf{W}^{(t')} - \mathbf{W}^*\|_F^2 \leqslant CLR^2 m^{-1},$$

where $C \geqslant 4$ is an absolute constant. Therefore, by triangle inequality, we further have the following for all $l \in [L]$,

$$\begin{aligned}
\|\mathbf{W}_l^{(t')} - \mathbf{W}_l^{(0)}\|_F &\leqslant \|\mathbf{W}_l^{(t')} - \mathbf{W}_l^*\|_F + \|\mathbf{W}_l^{(0)} - \mathbf{W}_l^*\|_F \\
&\leqslant \sqrt{CL}Rm^{-1/2} + Rm^{-1/2} \\
&\leqslant 2\sqrt{CL}Rm^{-1/2}. \tag{A.3}
\end{aligned}$$

Therefore, it is clear that $\|\mathbf{W}_l^{(t')} - \mathbf{W}_l^{(0)}\|_F \leqslant 2\sqrt{CL}Rm^{-1/2} \leqslant \tau$ based on our choice of $\tau$ previously. This completes the proof of the first part.

**Convergence of gradient descent.** (A.2) implies

$$\|\mathbf{W}^{(0)} - \mathbf{W}^*\|_F^2 - \|\mathbf{W}^{(T)} - \mathbf{W}^*\|_F^2 \geqslant \eta\bigg(\sum_{t=0}^{T-1} L_S(\mathbf{W}^{(t)}) - 2T\epsilon_{\text{NTRF}}\bigg).$$

Dividing by $\eta T$ on the both sides, we get

$$\frac{1}{T}\sum_{t=0}^{T-1} L_S(\mathbf{W}^{(t)}) \leqslant \frac{\|\mathbf{W}^{(0)} - \mathbf{W}^*\|_F^2}{\eta T} + 2\epsilon_{\text{NTRF}} \leqslant \frac{LR^2 m^{-1}}{\eta T} + 2\epsilon_{\text{NTRF}} \leqslant 3\epsilon_{\text{NTRF}},$$

where the second inequality is by the fact that $\mathbf{W}^* \in \mathcal{B}(\mathbf{W}^{(0)}, R \cdot m^{-1/2})$ and the last inequality is by our choices of $T$ and $\eta$ which ensure that $T\eta \geqslant LR^2 m^{-1}\epsilon_{\text{NTRF}}^{-1}$. Notice that $T = \lceil LR^2 m^{-1}\eta^{-1}\epsilon_{\text{NTRF}}^{-1}\rceil = \mathcal{O}(L^2 R^2 \epsilon_{\text{NTRF}}^{-1})$. This completes the proof of the second part, and we are able to complete the proof. □

## A.2 PROOF OF THEOREM 3.4

Following Cao and Gu (2020), we first introduce the definition of surrogate loss of the network, which is defined by the derivative of the loss function.

**Definition A.2.** We define the empirical surrogate error $\mathcal{E}_S(\mathbf{W})$ and population surrogate error $\mathcal{E}_\mathcal{D}(\mathbf{W})$ as follows:

$$\mathcal{E}_S(\mathbf{W}) := -\frac{1}{n}\sum_{i=1}^n \ell'\big[y_i \cdot f_\mathbf{W}(\boldsymbol{x}_i)\big], \ \ \mathcal{E}_\mathcal{D}(\mathbf{W}) := \mathbb{E}_{(\mathbf{x},y)\sim\mathcal{D}}\big\{ -\ell'\big[y \cdot f_\mathbf{W}(\mathbf{x})\big]\big\}.$$

The following lemma gives uniform-convergence type of results for $\mathcal{E}_S(\mathbf{W})$ utilizing the fact that $-\ell'(\cdot)$ is bounded and Lipschitz continuous.

**Lemma A.3.** For any $\widetilde{R}, \delta > 0$, suppose that $m = \widetilde{\Omega}(L^{12}\widetilde{R}^2) \cdot [\log(1/\delta)]^{3/2}$. Then with probability at least $1 - \delta$, it holds that

$$\big|\mathcal{E}_\mathcal{D}(\mathbf{W}) - \mathcal{E}_S(\mathbf{W})\big| \leqslant \widetilde{\mathcal{O}}\bigg(\min\bigg\{4^L L^{3/2}\widetilde{R}\sqrt{\frac{m}{n}}, \frac{L\widetilde{R}}{\sqrt{n}} + \frac{L^3\widetilde{R}^{4/3}}{m^{1/6}}\bigg\}\bigg) + \mathcal{O}\bigg(\sqrt{\frac{\log(1/\delta)}{n}}\bigg)$$

for all $\mathbf{W} \in \mathcal{B}(\mathbf{W}^{(0)}, \widetilde{R} \cdot m^{-1/2})$

We are now ready to prove Theorem 3.4, which combines the trajectory distance analysis in the proof of Theorem 3.3 with Lemma A.3.

*Proof of Theorem 3.4.* With exactly the same proof as Theorem 3.3, by (A.3) and induction we have $\mathbf{W}^{(0)}, \mathbf{W}^{(1)}, \ldots, \mathbf{W}^{(T)} \in \mathcal{B}(\mathbf{W}^{(0)}, \widetilde{R}m^{-1/2})$ with $\widetilde{R} = \mathcal{O}(\sqrt{L}R)$. Therefore by Lemma A.3, we have

$$\big|\mathcal{E}_\mathcal{D}(\mathbf{W}^{(t)}) - \mathcal{E}_S(\mathbf{W}^{(t)})\big| \leqslant \widetilde{\mathcal{O}}\bigg(\min\bigg\{4^L L^2 R\sqrt{\frac{m}{n}}, \frac{L^{3/2}R}{\sqrt{n}} + \frac{L^{11/3}R^{4/3}}{m^{1/6}}\bigg\}\bigg) + \mathcal{O}\bigg(\sqrt{\frac{\log(1/\delta)}{n}}\bigg)$$

for all $t = 0, 1, \ldots, T$. Note that we have $\mathbb{1}\{z < 0\} \leqslant -2\ell'(z)$. Therefore,

$$\mathbb{E}L_\mathcal{D}^{0-1}(\mathbf{W}^{(t)}) \leqslant 2\mathcal{E}_\mathcal{D}(\mathbf{W}^{(t)})$$

$$\leqslant 2L_S(\mathbf{W}^{(t)}) + \widetilde{\mathcal{O}}\bigg(\min\bigg\{4^L L^2 R\sqrt{\frac{m}{n}}, \frac{L^{3/2}R}{\sqrt{n}} + \frac{L^{11/3}R^{4/3}}{m^{1/6}}\bigg\}\bigg) + \mathcal{O}\bigg(\sqrt{\frac{\log(1/\delta)}{n}}\bigg)$$

for $t = 0, 1, \ldots, T$, where the last inequality is by $\mathcal{E}_S(\mathbf{W}) \leqslant L_S(\mathbf{W})$ because $-\ell'(z) \leqslant \ell(z)$ for all $z \in R$. This finishes the proof. □

## A.3 Proof of Theorem 3.5

In this section we provide the full proof of Theorem 3.5. We first give the following result, which is the counterpart of Lemma 5.1 for SGD. Again we pick $\mathbf{W}^* \in \mathcal{B}(\mathbf{W}^{(0)}, Rm^{-1/2})$ such that the loss of the corresponding NTRF model $F_{\mathbf{W}^{(0)}, \mathbf{W}^*}(\mathbf{x})$ achieves $\epsilon_{\text{NTRF}}$.

**Lemma A.4.** Set $\eta = \mathcal{O}(L^{-1}M(\tau)^{-2})$. Suppose that $\mathbf{W}^* \in \mathcal{B}(\mathbf{W}^{(0)}, \tau)$ and $\mathbf{W}^{(n')} \in \mathcal{B}(\mathbf{W}^{(0)}, \tau)$ for all $0 \leqslant n' \leqslant n - 1$. Then it holds that

$$\|\mathbf{W}^{(0)} - \mathbf{W}^*\|_F^2 - \|\mathbf{W}^{(n')} - \mathbf{W}^*\|_F^2 \geqslant \Big(\frac{3}{2} - 4\epsilon_{\text{app}}(\tau)\Big)\eta \sum_{i=1}^{n'} L_i(\mathbf{W}^{(i-1)}) - 2n\eta\epsilon_{\text{NTRF}}.$$

We introduce a surrogate loss $\mathcal{E}_i(\mathbf{W}) = -\ell'[y_i \cdot f_{\mathbf{W}}(\boldsymbol{x}_i)]$ and its population version $\mathcal{E}_{\mathcal{D}}(\mathbf{W}) = \mathbb{E}_{(\boldsymbol{x},y)\sim\mathcal{D}}[-\ell'[y \cdot f_{\mathbf{W}}(\boldsymbol{x})]]$, which have been used in (Ji and Telgarsky, 2018; Cao and Gu, 2019; Ji and Telgarsky, 2020). Our proof is based on the application of Lemma A.4 and an online-to-batch conversion argument (Cesa-Bianchi et al., 2004; Cao and Gu, 2019; Ji and Telgarsky, 2020). We introduce a surrogate loss $\mathcal{E}_i(\mathbf{W}) = -\ell'[y_i \cdot f_{\mathbf{W}}(\boldsymbol{x}_i)]$ and its population version $\mathcal{E}_{\mathcal{D}}(\mathbf{W}) = \mathbb{E}_{(\boldsymbol{x},y)\sim\mathcal{D}}[-\ell'(y \cdot f_{\mathbf{W}}(\boldsymbol{x}))]$, which have been used in (Ji and Telgarsky, 2018; Cao and Gu, 2019; Nitanda and Suzuki, 2019; Ji and Telgarsky, 2020).

*Proof of Theorem 3.5.* Recall that $\mathbf{W}^*$ is chosen such that

$$\frac{1}{n}\sum_{i=1}^{n} \ell\big(y_i F_{\mathbf{W}^{(0)}, \mathbf{W}^*}(\mathbf{x}_i)\big) = \epsilon_{\text{NTRF}}$$

and $\mathbf{W}^* \in \mathcal{B}(\mathbf{W}^{(0)}, Rm^{-1/2})$. To apply Lemma A.4, we need the region $\mathcal{B}(\mathbf{W}^{(0)}, \tau)$ to include both $\mathbf{W}^*$ and $\{\mathbf{W}^{(t)}\}_{t=0,\dots,t'}$. This motivates us to set $\tau = \widetilde{\mathcal{O}}(L^{1/2}m^{-1/2}R)$, which is slightly larger than $m^{-1/2}R$. With this choice of $\tau$, by Lemma A.1 we have $\epsilon_{\text{app}}(\tau) = \widetilde{\mathcal{O}}(\tau^{4/3}m^{1/2}L^3) = \widetilde{\mathcal{O}}(R^{4/3}L^{11/3}m^{-1/6})$. Therefore, we can set

$$m = \widetilde{\Omega}(R^8 L^{22})$$

to ensure that $\epsilon_{\text{app}}(\tau) \leqslant 1/8$, where $\widetilde{\Omega}(\cdot)$ hides polylogarithmic dependencies on network depth $L$, NTRF function class size $R$, and failure probability parameter $\delta$.

Then by Lemma A.4, we have with probability at least $1 - \delta$,

$$\|\mathbf{W}^{(0)} - \mathbf{W}^*\|_F^2 - \|\mathbf{W}^{(n')} - \mathbf{W}^*\|_F^2 \geqslant \eta \sum_{i=1}^{n'} L_i(\mathbf{W}^{(i-1)}) - 2n\eta\epsilon_{\text{NTRF}} \qquad (A.4)$$

as long as $\mathbf{W}^{(0)}, \dots, \mathbf{W}^{(n'-1)} \in \mathcal{B}(\mathbf{W}^{(0)}, \tau)$.

We then prove Theorem 3.5 in two steps: 1) all iterates stay inside $\mathcal{B}(\mathbf{W}^{(0)}, \tau)$; and 2) convergence of online SGD.

**All iterates stay inside $\mathcal{B}(\mathbf{W}^{(0)}, \tau)$.** Similar to the proof of Theorem 3.3, we prove this part by induction. Assuming $\mathbf{W}^{(i)}$ satisfies $\mathbf{W}^{(i)} \in \mathcal{B}(\mathbf{W}^{(0)}, \tau)$ for all $i \leqslant n' - 1$, by (A.4), we have

$$\|\mathbf{W}^{(n')} - \mathbf{W}^*\|_F^2 \leqslant \|\mathbf{W}^{(0)} - \mathbf{W}^*\|_F^2 + 2n\eta\epsilon_{\text{NTRF}}$$
$$\leqslant LR^2 \cdot m^{-1} + 2n\eta\epsilon_{\text{NTRF}},$$

where the last inequality is by $\mathbf{W}^* \in \mathcal{B}(\mathbf{W}^{(0)}, Rm^{-1/2})$. Then by triangle inequality, we further get

$$\|\mathbf{W}_l^{(n')} - \mathbf{W}_l^{(0)}\|_F \leqslant \|\mathbf{W}_l^{(n')} - \mathbf{W}_l^*\|_F + \|\mathbf{W}_l^* - \mathbf{W}_l^{(0)}\|_F$$
$$\leqslant \|\mathbf{W}^{(n')} - \mathbf{W}^*\|_F + \|\mathbf{W}_l^* - \mathbf{W}_l^{(0)}\|_F$$
$$\leqslant \mathcal{O}(\sqrt{L}Rm^{-1/2} + \sqrt{n\eta\epsilon_{\text{NTRF}}}).$$

Then by our choices of $\eta = \Theta\big(m^{-1} \cdot (LR^2 n^{-1}\epsilon_{\text{NTRF}}^{-1} \wedge L^{-1})\big)$, we have $\|\mathbf{W}^{(n')} - \mathbf{W}^{(0)}\|_F \leqslant 2\sqrt{L}Rm^{-1/2} \leqslant \tau$. This completes the proof of the first part.

**Convergence of online SGD.** By (A.4), we have

$$\|\mathbf{W}^{(0)} - \mathbf{W}^*\|_F^2 - \|\mathbf{W}^{(n)} - \mathbf{W}^*\|_F^2 \geqslant \eta\left( \sum_{i=1}^n L_i(\mathbf{W}^{(i-1)}) - 2n\epsilon_{\mathrm{NTRF}} \right).$$

Dividing by $\eta n$ on the both sides and rearranging terms, we get

$$\frac{1}{n} \sum_{i=1}^n L_i(\mathbf{W}^{(i-1)}) \leqslant \frac{\|\mathbf{W}^{(0)} - \mathbf{W}^*\|_F^2 - \|\mathbf{W}^{(n)} - \mathbf{W}^*\|_F^2}{\eta n} + 2\epsilon_{\mathrm{NTRF}} \leqslant \frac{L^2 R^2}{n} + 3\epsilon_{\mathrm{NTRF}},$$

where the second inequality follows from facts that $\mathbf{W}^* \in \mathcal{B}(\mathbf{W}^{(0)}, R \cdot m^{-1/2})$ and $\eta = \Theta\big(m^{-1} \cdot (LR^2 n^{-1}\epsilon_{\mathrm{NTRF}}^{-1} \wedge L^{-1})\big)$. By Lemma 4.3 in (Ji and Telgarsky, 2020) and the fact that $\mathcal{E}_i(\mathbf{W}^{(i-1)}) \leqslant L_i(\mathbf{W}^{(i-1)})$, we have

$$\begin{aligned}
\frac{1}{n} \sum_{i=1}^n L_{\mathcal{D}}^{0-1}(\mathbf{W}^{(i-1)}) &\leqslant \frac{2}{n} \sum_{i=1}^n \mathcal{E}_{\mathcal{D}}(\mathbf{W}^{(i-1)}) \\
&\leqslant \frac{8}{n} \sum_{i=1}^n \mathcal{E}_i(\mathbf{W}^{(i-1)}) + \frac{8\log(1/\delta)}{n} \\
&\leqslant \frac{8L^2 R^2}{n} + \frac{8\log(1/\delta)}{n} + 24\epsilon_{\mathrm{NTRF}}.
\end{aligned}$$

This completes the proof of the second part. $\qquad\square$

# B   PROOF OF RESULTS IN SECTION 4

## B.1   PROOF OF PROPOSITION 4.2

We first provide the following lemma which gives an upper bound of the neural network output at the initialization.

**Lemma B.1** (Lemma 4.4 in Cao and Gu (2019)). *Under Assumption 3.1, if $m \geqslant \bar{C}L\log(nL/\delta)$ with some absolute constant $\bar{C}$, with probability at least $1 - \delta$, we have*

$$|f_{\mathbf{W}^{(0)}}(\mathbf{x}_i)| \leqslant C\sqrt{\log(n/\delta)}$$

*for some absolute constant $C$.*

*Proof of Proposition 4.2.* Under Assumption 4.1, we can find a collection of matrices $\mathbf{U}^* = \{\mathbf{U}_1^*, \cdots, \mathbf{U}_L^*\}$ with $\sum_{l=1}^L \|\mathbf{U}_l^*\|_F^2 = 1$ such that $y_i\langle \nabla f_{\mathbf{W}^{(0)}}(\mathbf{x}_i), \mathbf{U}^* \rangle \geqslant m^{1/2}\gamma$ for at least $1 - \rho$ fraction of the training data. By Lemma B.1, for all $i \in [n]$ we have $|f_{\mathbf{W}^{(0)}}(\mathbf{x}_i)| \leqslant C\sqrt{\log(n/\delta)}$ for some absolute constant $C$. Then for any positive constant $\lambda$, we have for at least $1 - \rho$ portion of the data,

$$y_i\big(f_{\mathbf{W}^{(0)}}(\mathbf{x}_i) + \langle \nabla f_{\mathbf{W}^{(0)}}, \lambda\mathbf{U}^* \rangle\big) \geqslant m^{1/2}\lambda\gamma - C\sqrt{\log(n/\delta)}.$$

For this fraction of data, we can set

$$\lambda = \frac{C'\big[\log^{1/2}(n/\delta) + \log(1/\epsilon)\big]}{m^{1/2}\gamma},$$

where $C'$ is an absolute constant, and get

$$m^{1/2}\lambda\gamma - C\sqrt{\log(n/\delta)} \geqslant \log(1/\epsilon).$$

Now we let $\mathbf{W}^* = \mathbf{W}^{(0)} + \lambda\mathbf{U}^*$. By the choice of $R$ in Proposition 4.2, we have $\mathbf{W}^* \in \mathcal{B}(\mathbf{W}^{(0)}, R \cdot m^{-1/2})$. The above inequality implies that for at least $1 - \rho$ fraction of data, we have $\ell\big(y_i F_{\mathbf{W}^{(0)}, \mathbf{W}^*}(\mathbf{x}_i)\big) \leqslant \epsilon$. For the rest data, we have

$$y_i\big(f_{\mathbf{W}^{(0)}}(\mathbf{x}_i) + \langle \nabla f_{\mathbf{W}^{(0)}}, \lambda\mathbf{U}^* \rangle\big) \geqslant -C\sqrt{\log(n/\delta)} - \lambda\|\nabla f_{\mathbf{W}^{(0)}}\|_2^2 \geqslant -C_1 R$$

for some absolute positive constant $C_1$, where the last inequality follows from fact that $\|\nabla f_{\mathbf{W}^{(0)}}\|_2 = \tilde{\mathcal{O}}(m^{1/2})$ (see Lemma A.1 for detail). Then note that we use cross-entropy loss, it follows that for this fraction of training data, we have $\ell\big(y_i F_{\mathbf{W}^{(0)},\mathbf{W}*}(\mathbf{x}_i)\big) \leqslant C_2 R$ for some constant $C_2$. Combining the results of these two fractions of training data, we can conclude

$$\epsilon_{\text{NTRF}} \leqslant n^{-1} \sum_{i=1}^{n} \ell\big(y_i F_{\mathbf{W}^{(0)},\mathbf{W}*}(\mathbf{x}_i)\big) \leqslant (1-\rho)\epsilon + \rho \cdot \mathcal{O}(R)$$

This completes the proof.

$\square$

## B.2 PROOF OF PROPOSITION 4.4

*Proof of Proposition 4.4.* We are going to prove that Assumption 4.3 implies the existence of a good function in the NTRF function class.

By Definition 3.2 and the definition of cross-entropy loss, our goal is to prove that there exists a collection of matrices $\overline{\mathbf{W}} = \{\overline{\mathbf{W}}_1, \overline{\mathbf{W}}_2\}$ satisfying $\max\{\|\overline{\mathbf{W}}_1 - \mathbf{W}_1^{(0)}\|_F, \|\overline{\mathbf{W}}_2 - \mathbf{W}_2^{(0)}\|_2\} \leqslant R \cdot m^{-1/2}$ such that

$$y_i \cdot \big[f_{\mathbf{W}^{(0)}}(\mathbf{x}_i) + \langle \nabla_{\mathbf{W}_1} f_{\mathbf{W}^{(0)}}, \overline{\mathbf{W}}_1 - \mathbf{W}_1^{(0)} \rangle + \langle \nabla_{\mathbf{W}_2} f_{\mathbf{W}^{(0)}}, \overline{\mathbf{W}}_2 - \mathbf{W}_2^{(0)} \rangle\big] \geqslant \log(2/\epsilon).$$

We first consider $\nabla_{\mathbf{W}_1} f_{\mathbf{W}^{(0)}}(\mathbf{x}_i)$, which has the form

$$\big(\nabla_{\mathbf{W}_1} f_{\mathbf{W}^{(0)}}(\mathbf{x}_i)\big)_j = m^{1/2} \cdot w_{2,j}^{(0)} \cdot \sigma'(\langle \mathbf{w}_{1,j}^{(0)}, \mathbf{x}_i \rangle) \cdot \mathbf{x}_i.$$

Note that $w_{2,j}^{(0)}$ and $\mathbf{w}_{1,j}^{(0)}$ are independently generated from $\mathcal{N}(0, 1/m)$ and $\mathcal{N}(0, 2\mathbf{I}/m)$ respectively, thus we have $\mathbb{P}(|w_{2,j}^{(0)}| \geqslant 0.47m^{-1/2}) \geqslant 1/2$. By Hoeffeding's inequality, we know that with probability at least $1 - \exp(-m/8)$, there are at least $m/4$ nodes, whose union is denoted by $\mathcal{S}$, satisfying $|w_{2,j}^{(0)}| \geqslant 0.47m^{-1/2}$. Then we only focus on the nodes in the set $\mathcal{S}$. Note that $\mathbf{W}_1^{(0)}$ and $\mathbf{W}_2^{(0)}$ are independently generated. Then by Assumption 4.3 and Hoeffeding's inequality, there exists a function $\overline{\mathbf{u}}(\cdot) : \mathbb{R}^d \to \mathbb{R}^d$ such that with probability at least $1 - \delta'$,

$$\frac{1}{|\mathcal{S}|} \sum_{j \in \mathcal{S}} y_i \cdot \langle \overline{\mathbf{u}}(\mathbf{w}_{1,j}^{(0)}), \mathbf{x}_i \rangle \cdot \sigma'(\langle \mathbf{w}_{1,j}^{(0)}, \mathbf{x}_i \rangle) \geqslant \gamma - \sqrt{\frac{2\log(1/\delta')}{|\mathcal{S}|}}.$$

Define $\mathbf{v}_j = \overline{\mathbf{u}}(\mathbf{w}_{1,j}^{(0)})/w_{2,j}$ if $|w_{2,j}| \geqslant 0.47m^{-1/2}$ and $\mathbf{v}_j = \mathbf{0}$ otherwise. Then we have

$$\sum_{j=1}^{m} y_i \cdot w_{2,j}^{(0)} \cdot \langle \mathbf{v}_j, \mathbf{x}_i \rangle \cdot \sigma'(\langle \mathbf{w}_{1,j}^{(0)}, \mathbf{x}_i \rangle) = \sum_{j \in \mathcal{S}} y_i \cdot \langle \overline{\mathbf{u}}(\mathbf{w}_{1,j}^{(0)}), \mathbf{x}_i \rangle \cdot \sigma'(\langle \mathbf{w}_{1,j}^{(0)}, \mathbf{x}_i \rangle)$$

$$\geqslant |\mathcal{S}|\gamma - \sqrt{2|\mathcal{S}|\log(1/\delta')}.$$

Set $\delta = 2n\delta'$ and apply union bound, we have with probability at least $1 - \delta/2$,

$$\sum_{j=1}^{m} y_i \cdot w_{2,j}^{(0)} \cdot \langle \mathbf{v}_j, \mathbf{x}_i \rangle \cdot \sigma'(\langle \mathbf{w}_{1,j}^{(0)}, \mathbf{x}_i \rangle) \geqslant |\mathcal{S}|\gamma - \sqrt{2|\mathcal{S}|\log(2n/\delta)}.$$

Therefore, note that with probability at least $1 - \exp(-m/8)$, we have $|\mathcal{S}| \geqslant m/4$. Moreover, in Assumption 4.3, by $y_i \in \{\pm 1\}$ and $|\sigma'(\cdot)|, \|\overline{\mathbf{u}}(\cdot)\|_2, \|\mathbf{x}_i\|_2 \leqslant 1$ for $i = 1, \ldots, n$, we see that $\gamma \leqslant 1$. Then if $m \geqslant 32\log(n/\delta)/\gamma^2$, with probability at least $1 - \delta/2 - \exp\big(-4\log(n/\delta)/\gamma^2\big) \geqslant 1 - \delta$,

$$\sum_{j=1}^{m} y_i \cdot w_{2,j}^{(0)} \cdot \langle \mathbf{v}_j, \mathbf{x}_i \rangle \cdot \sigma'(\langle \mathbf{w}_{1,j}^{(0)}, \mathbf{x}_i \rangle) \geqslant |\mathcal{S}|\gamma/2.$$

Let $\mathbf{U} = (\mathbf{v}_1, \mathbf{v}_2, \cdots, \mathbf{v}_m)^\top/\sqrt{m|\mathcal{S}|}$, we have

$$y_i \langle \nabla_{\mathbf{W}_1} f_{\mathbf{W}^{(0)}}(\mathbf{x}_i), \mathbf{U} \rangle = \frac{1}{\sqrt{|\mathcal{S}|}} \sum_{j=1}^{m} y_i \cdot w_{2,j}^{(0)} \cdot \langle \mathbf{v}_j, \mathbf{x}_i \rangle \cdot \sigma'(\langle \mathbf{w}_{1,j}^{(0)}, \mathbf{x}_i \rangle) \geqslant \frac{\sqrt{|\mathcal{S}|}\gamma}{2} \geqslant \frac{m^{1/2}\gamma}{4},$$

where the last inequality is by the fact that $|\mathcal{S}| \geqslant m/4$. Besides, note that by concentration and Gaussian tail bound, we have $|f_{\mathbf{W}^{(0)}}(\mathbf{x}_i)| \leqslant C\log(n/\delta)$ for some absolute constant $C$. Therefore, let $\overline{\mathbf{W}}_1 = \mathbf{W}_1^{(0)} + 4\big(\log(2/\epsilon) + C\log(n/\delta)\big)m^{-1/2}\mathbf{U}/\gamma$ and $\overline{\mathbf{W}}_2 = \mathbf{W}_2^{(0)}$, we have

$$y_i \cdot \big[f_{\mathbf{W}^{(0)}}(\mathbf{x}_i) + \langle\nabla_{\mathbf{W}_1}f_{\mathbf{W}^{(0)}}, \overline{\mathbf{W}}_1 - \mathbf{W}_1^{(0)}\rangle + \langle\nabla_{\mathbf{W}_2}f_{\mathbf{W}^{(0)}}, \overline{\mathbf{W}}_2 - \mathbf{W}_2^{(0)}\rangle\big] \geqslant \log(2/\epsilon). \quad \text{(B.1)}$$

Note that $\|\overline{\mathbf{u}}(\cdot)\|_2 \leqslant 1$, we have $\|\mathbf{U}\|_F \leqslant 1/0.47 \leqslant 2.2$. Therefore, we further have $\|\overline{\mathbf{W}}_1 - \mathbf{W}_1^{(0)}\|_F \leqslant 8.8\gamma^{-1}\big(\log(2/\epsilon) + C\log(n/\delta)\big) \cdot m^{-1/2}$. This implies that $\overline{\mathbf{W}} \in \mathcal{B}(\mathbf{W}^{(0)}, R)$ with $R = \mathcal{O}\big(\log\big(n/(\delta\epsilon)\big)/\gamma\big)$. Applying the inequality $\ell(\log(2/\epsilon)) \leqslant \epsilon$ on (B.1) gives

$$\ell(y_i \cdot F_{\mathbf{W}^{(0)}, \overline{\mathbf{W}}}(\mathbf{x}_i)) \leqslant \epsilon$$

for all $i = 1, \ldots, n$. This completes the proof. $\qquad\square$

## B.3 Proof of Proposition 4.6

Based on our theoretical analysis, the major goal is to show that there exist certain choices of $R$ and $m$ such that the best NTRF model in the function class $\mathcal{F}(\mathbf{W}^{(0)}, R)$ can achieve $\epsilon$ training error. In this proof, we will prove a stronger results by showing that given the quantities of $R$ and $m$ specified in Proposition 4.6, there exists a NTRF model with parameter $\mathbf{W}^*$ that satisfies $n^{-1}\sum_{i=1}^n \ell\big(y_i F_{\mathbf{W}^{(0)}, \mathbf{W}^*}(\mathbf{x}_i)\big) \leqslant \epsilon$.

In order to do so, we consider training the NTRF model via a different surrogate loss function. Specifically, we consider squared hinge loss $\widetilde{\ell}(x) = \big(\max\{\lambda - x, 0\}\big)^2$, where $\lambda$ denotes the target margin. In the later proof, we choose $\lambda = \log(1/\epsilon) + 1$ such that the condition $\widetilde{\ell}(x) \leqslant 1$ can guarantee that $x \geqslant \log(\epsilon)$. Moreover, we consider using gradient flow, i.e., gradient descent with infinitesimal step size, to train the NTRF model. Therefore, in the remaining part of the proof, we consider optimizing the NTRF parameter $\mathbf{W}$ with the loss function

$$\widetilde{L}_S(\mathbf{W}) = \frac{1}{n}\sum_{i=1}^n \widetilde{\ell}\big(y_i F_{\mathbf{W}^{(0)}, \mathbf{W}}(\mathbf{x}_i)\big).$$

Moreover, for simplicity, we only consider optimizing parameter in the last hidden layer (i.e., $\mathbf{W}_{L-1}$). Then the gradient flow can be formulated as

$$\frac{\mathrm{d}\mathbf{W}_{L-1}(t)}{\mathrm{d}t} = -\nabla_{\mathbf{W}_{L-1}}\widetilde{L}_S(\mathbf{W}(t)), \quad \frac{\mathrm{d}\mathbf{W}_l(t)}{\mathrm{d}t} = \mathbf{0} \quad \text{for any } l \neq L-1.$$

Note that the NTRF model is a linear model, thus by Definition 3.2, we have

$$\nabla_{\mathbf{W}_{L-1}}\widetilde{L}_S(\mathbf{W}(t)) = y_i\widetilde{\ell}'\big(y_i F_{\mathbf{W}^{(0)}, \mathbf{W}(t)}(\mathbf{x}_i)\big) \cdot \nabla_{\mathbf{W}_{L-1}}F_{\mathbf{W}^{(0)}, \mathbf{W}(t)}(\mathbf{x}_i)$$
$$= y_i\widetilde{\ell}'\big(y_i F_{\mathbf{W}^{(0)}, \mathbf{W}(t)}(\mathbf{x}_i)\big) \cdot \nabla_{\mathbf{W}_{L-1}^{(0)}}f_{\mathbf{W}^{(0)}}(\mathbf{x}_i). \quad \text{(B.2)}$$

Then it is clear that $\nabla_{\mathbf{W}_{L-1}}\widetilde{L}_S(\mathbf{W}(t))$ has fixed direction throughout the optimization.

In order to prove the convergence of gradient flow and characterize the quantity of $R$, We first provide the following lemma which gives an upper bound of the NTRF model output at the initialization.

Then we provide the following lemma which characterizes a lower bound of the Frobenius norm of the partial gradient $\nabla_{\mathbf{W}_{L-1}}\widetilde{L}_S(\mathbf{W})$.

**Lemma B.2** (Lemma B.5 in Zou et al. (2019)). *Under Assumptions 3.1 and 4.5, if $m = \widetilde{\Omega}(n^2\phi^{-1})$, then for all $t \geqslant 0$, with probability at least $1 - \exp\big(-O(m\phi/n)\big)$, there exist a positive constant $C$ such that*

$$\|\nabla_{\mathbf{W}_{L-1}}\widetilde{L}_S(\mathbf{W}(t))\|_F^2 \geqslant \frac{Cm\phi}{n^5}\bigg[\sum_{i=1}^n \widetilde{\ell}'\big(y_i F_{\mathbf{W}^{(0)}, \mathbf{W}(t)}(\mathbf{x}_i)\big)\bigg]^2.$$

We slightly modified the original version of this lemma since we use different models (we consider NTRF model while Zou et al. (2019) considers neural network model). However, by (B.2), it is clear that the gradient $\nabla\widetilde{L}_S(\mathbf{W})$ can be regarded as a type of the gradient for neural network model at the initialization (i.e., $\nabla_{\mathbf{W}_{L-1}}L_S(\mathbf{W}^{(0)})$) is valid. Now we are ready to present the proof.

*Proof of Proposition 4.6.* Recall that we only consider training the last hidden weights, i.e., $\mathbf{W}_{L-1}$, via gradient flow with squared hinge loss, and our goal is to prove that gradient flow is able to find a NTRF model within the function class $\mathcal{F}(\mathbf{W}^{(0)}, R)$ around the initialization, i.e., achieving $n^{-1} \sum_{i=1}^{n} \ell(y_i F_{\mathbf{W}^{(0)}, \mathbf{W}*}(\mathbf{x}_i)) \leqslant \epsilon$. Let $\mathbf{W}(t)$ be the weights at time $t$, gradient flow implies that

$$\frac{\mathrm{d}\widetilde{L}_S(\mathbf{W}(t))}{\mathrm{d}t} = -\|\nabla_{\mathbf{W}_{L-1}}\widetilde{L}_S(\mathbf{W}(t))\|_F^2 \leqslant -\frac{Cm\phi}{n^5}\bigg(\sum_{i=1}^{n}\widetilde{\ell}'(y_i F_{\mathbf{W}^{(0)}, \mathbf{W}(t)}(\mathbf{x}_i))\bigg)^2 = \frac{4Cm\phi\widetilde{L}_S(\mathbf{W}(t))}{n^3},$$

where the first equality is due to the fact that we only train the last hidden layer, the first inequality is by Lemma B.2 and the second equality follows from the fact that $\widetilde{\ell}'(\cdot) = -2\sqrt{\widetilde{\ell}(\cdot)}$. Solving the above inequality gives

$$\widetilde{L}_S(\mathbf{W}(t)) \leqslant \widetilde{L}_S(\mathbf{W}(0)) \cdot \exp\bigg(-\frac{4Cm\phi t}{n^3}\bigg). \tag{B.3}$$

Then, set $T = \mathcal{O}(n^3 m^{-1} \phi^{-1} \cdot \log(\widetilde{L}_S(\mathbf{W}(0))/\epsilon'))$ and $\epsilon' = 1/n$, we have $\widetilde{L}_S(\mathbf{W}(t)) \leqslant \epsilon'$. Then it follows that $\widetilde{\ell}(y_i F_{\mathbf{W}^{(0)}, \mathbf{W}(t)}(\mathbf{x}_i)) \leqslant 1$, which implies that $y_i F_{\mathbf{W}^{(0)}, \mathbf{W}(t)}(\mathbf{x}_i) \geqslant \log(\epsilon)$ and thus $n^{-1} \sum_{i=1}^{n} \ell(y_i F_{\mathbf{W}^{(0)}, \mathbf{W}*}(\mathbf{x}_i)) \leqslant \epsilon$. Therefore, $\mathbf{W}(T)$ is exactly the NTRF model we are looking for.

The next step is to characterize the distance between $\mathbf{W}(T)$ and $\mathbf{W}(0)$ in order to characterize the quantity of $R$. Note that $\|\nabla_{\mathbf{W}_{L-1}}\widetilde{L}_S(\mathbf{W}(t))\|_F^2 \geqslant 4Cm\phi\widetilde{L}_S(\mathbf{W}(t))/n^3$, we have

$$\frac{\mathrm{d}\sqrt{\widetilde{L}_S(\mathbf{W}(t))}}{\mathrm{d}t} = -\frac{\|\nabla_{\mathbf{W}_{L-1}}\widetilde{L}_S(\mathbf{W}(t))\|_F^2}{2\sqrt{\widetilde{L}_S(\mathbf{W}(t))}} \leqslant -\|\nabla_{\mathbf{W}_{L-1}}\widetilde{L}_S(\mathbf{W}(t))\|_F \cdot \frac{C^{1/2}m^{1/2}\phi^{1/2}}{n^{3/2}}.$$

Taking integral on both sides and rearranging terms, we have

$$\int_{t=0}^{T} \|\nabla_{\mathbf{W}_{L-1}}\widetilde{L}_S(\mathbf{W}(t))\|_F \mathrm{d}t \leqslant \frac{n^{3/2}}{C^{1/2}m^{1/2}\phi^{1/2}} \cdot \bigg(\sqrt{\widetilde{L}_S(\mathbf{W}(0))} - \sqrt{\widetilde{L}_S(\mathbf{W}(t))}\bigg).$$

Note that the L.H.S. of the above inequality is an upper bound of $\|\mathbf{W}(t) - \mathbf{W}(0)\|_F$, we have for any $t \geqslant 0$,

$$\|\mathbf{W}(t) - \mathbf{W}(0)\|_F \leqslant \frac{n^{3/2}}{C^{1/2}m^{1/2}\phi^{1/2}} \cdot \sqrt{\widetilde{L}_S(\mathbf{W}(0))} = \mathcal{O}\bigg(\frac{n^{3/2}\log(n/(\delta\epsilon))}{m^{1/2}\phi^{1/2}}\bigg),$$

where the second inequality is by Lemma B.1 and our choice of $\lambda = \log(1/\epsilon) + 1$. This implies that there exists a point $\mathbf{W}*$ within the class $\mathcal{F}(\mathbf{W}^{(0)}, R)$ with

$$R = \mathcal{O}\bigg(\frac{n^{3/2}\log(n/(\delta\epsilon))}{\phi^{1/2}}\bigg)$$

such that

$$\epsilon_{\mathrm{NTRF}} := n^{-1} \sum_{i=1}^{n} \ell(y_i F_{\mathbf{W}^{(0)}, \mathbf{W}*}(\mathbf{x}_i)) \leqslant \epsilon.$$

Then by Theorem 3.3, and, more specifically, (A.1), we can compute the minimal required neural network width as follows,

$$m = \widetilde{\Omega}(R^8 L^{22}) = \widetilde{\Omega}\bigg(\frac{L^{22}n^{12}}{\phi^4}\bigg).$$

This completes the proof. □

## C    PROOF OF TECHNICAL LEMMAS

Here we provide the proof of Lemmas 5.1, A.3 and A.4.

## C.1 PROOF OF LEMMA 5.1

The detailed proof of Lemma 5.1 is given as follows.

*Proof of Lemma 5.1.* Based on the update rule of gradient descent, i.e., $\mathbf{W}^{(t+1)} = \mathbf{W}^{(t)} - \eta \nabla_{\mathbf{W}} L_S(\mathbf{W}^{(t)})$, we have the following calculation.

$$
\|\mathbf{W}^{(t)} - \mathbf{W}^*\|_F^2 - \|\mathbf{W}^{(t+1)} - \mathbf{W}^*\|_F^2
$$
$$
= \underbrace{\frac{2\eta}{n} \sum_{i=1}^{n} \langle \mathbf{W}^{(t)} - \mathbf{W}^*, \nabla_{\mathbf{W}} L_i(\mathbf{W}^{(t)}) \rangle}_{I_1} - \underbrace{\eta^2 \sum_{l=1}^{L} \|\nabla_{\mathbf{W}_l} L_S(\mathbf{W}^{(t)})\|_F^2}_{I_2}, \quad \text{(C.1)}
$$

where the equation follows from the fact that $L_S(\mathbf{W}^{(t)}) = n^{-1} \sum_{i=1}^{n} L_i(\mathbf{W}^{(t)})$. In what follows, we first bound the term $I_1$ on the R.H.S. of (C.1) by approximating the neural network functions with linear models. By assumption, for $t = 0, \ldots, t' - 1$, $\mathbf{W}^{(t)}, \mathbf{W}^* \in \mathcal{B}(\mathbf{W}^{(0)}, \tau)$. Therefore by the definition of $\epsilon_{\mathrm{app}}(\tau)$,

$$
y_i \cdot \langle \nabla f_{\mathbf{W}^{(t)}}(\mathbf{x}_i), \mathbf{W}^{(t)} - \mathbf{W}^* \rangle \leqslant y_i \cdot \big( f_{\mathbf{W}^{(t)}}(\mathbf{x}_i) - f_{\mathbf{W}^*}(\mathbf{x}_i) \big) + \epsilon_{\mathrm{app}}(\tau) \quad \text{(C.2)}
$$

Moreover, we also have

$$
0 \leqslant y_i \cdot \big( f_{\mathbf{W}^*}(\mathbf{x}_i) - f_{\mathbf{W}^{(0)}}(\mathbf{x}_i) - \langle \nabla f_{\mathbf{W}^{(0)}}(\mathbf{x}_i), \mathbf{W}^* - \mathbf{W}^{(0)} \rangle \big) + \epsilon_{\mathrm{app}}(\tau)
$$
$$
= y_i \cdot \big( f_{\mathbf{W}^*}(\mathbf{x}_i) - F_{\mathbf{W}^{(0)}, \mathbf{W}^*}(\mathbf{x}_i) \big) + \epsilon_{\mathrm{app}}(\tau), \quad \text{(C.3)}
$$

where the equation follows by the definition of $F_{\mathbf{W}^{(0)}, \mathbf{W}^*}(\mathbf{x})$. Adding (C.3) to (C.2) and canceling the terms $y_i \cdot f_{\mathbf{W}^*}(\mathbf{x}_i)$, we obtain that

$$
y_i \cdot \langle \nabla f_{\mathbf{W}^{(t)}}(\mathbf{x}_i), \mathbf{W}^{(t)} - \mathbf{W}^* \rangle \leqslant y_i \cdot \big( f_{\mathbf{W}^{(t)}}(\mathbf{x}_i) - F_{\mathbf{W}^{(0)}, \mathbf{W}^*}(\mathbf{x}_i) \big) + 2\epsilon_{\mathrm{app}}(\tau). \quad \text{(C.4)}
$$

We can now give a lower bound on first term on the R.H.S. of (C.1). For $i = 1, \ldots, n$, applying the chain rule on the loss function gradients and utilizing (C.4), we have

$$
\langle \mathbf{W}^{(t)} - \mathbf{W}^*, \nabla_{\mathbf{W}} L_i(\mathbf{W}^{(t)}) \rangle = \ell'\big(y_i f_{\mathbf{W}^{(t)}}(\mathbf{x}_i)\big) \cdot y_i \cdot \langle \mathbf{W}^{(t)} - \mathbf{W}^*, \nabla_{\mathbf{W}} f_{\mathbf{W}^{(t)}}(\mathbf{x}_i) \rangle
$$
$$
\geqslant \ell'\big(y_i f_{\mathbf{W}^{(t)}}(\mathbf{x}_i)\big) \cdot \big(y_i f_{\mathbf{W}^{(t)}}(\mathbf{x}_i) - y_i f_{\mathbf{W}^*}(\mathbf{x}_i) + 2\epsilon_{\mathrm{app}}(\tau)\big)
$$
$$
\geqslant (1 - 2\epsilon_{\mathrm{app}}(\tau)) \ell\big(y_i f_{\mathbf{W}^{(t)}}(\mathbf{x}_i)\big) - \ell\big(y_i F_{\mathbf{W}^{(0)}, \mathbf{W}^*}(\mathbf{x}_i)\big), \quad \text{(C.5)}
$$

where the first inequality is by the fact that $\ell'\big(y_i f_{\mathbf{W}^{(t)}}(\mathbf{x}_i)\big) < 0$, the second inequality is by convexity of $\ell(\cdot)$ and the fact that $-\ell'\big(y_i f_{\mathbf{W}^{(t)}}(\mathbf{x}_i)\big) \leqslant \ell\big(y_i f_{\mathbf{W}^{(t)}}(\mathbf{x}_i)\big)$.

We now proceed to bound the term $I_2$ on the R.H.S. of (C.1). Note that we have $\ell'(\cdot) < 0$, and therefore the Frobenius norm of the gradient $\nabla_{\mathbf{W}_l} L_S(\mathbf{W}^{(t)})$ can be upper bounded as follows,

$$
\|\nabla_{\mathbf{W}_l} L_S(\mathbf{W}^{(t)})\|_F = \left\| \frac{1}{n} \sum_{i=1}^{n} \ell'\big(y_i f_{\mathbf{W}^{(t)}}(\mathbf{x}_i)\big) \nabla_{\mathbf{W}_l} f_{\mathbf{W}^{(t)}}(\mathbf{x}_i) \right\|_F
$$
$$
\leqslant \frac{1}{n} \sum_{i=1}^{n} -\ell'\big(y_i f_{\mathbf{W}^{(t)}}(\mathbf{x}_i)\big) \cdot \|\nabla_{\mathbf{W}_l} f_{\mathbf{W}^{(t)}}(\mathbf{x}_i)\|_F,
$$

where the inequality follows by triangle inequality. We now utilize the fact that cross-entropy loss satisfies the inequalities $-\ell'(\cdot) \leqslant \ell(\cdot)$ and $-\ell'(\cdot) \leqslant 1$. Therefore by definition of $M(\tau)$, we have

$$
\sum_{l=1}^{L} \|\nabla_{\mathbf{W}_l} L_S(\mathbf{W}^{(t)})\|_F^2 \leqslant \mathcal{O}\big(LM(\tau)^2\big) \cdot \left( \frac{1}{n} \sum_{i=1}^{n} -\ell'\big(y_i f_{\mathbf{W}^{(t)}}(\mathbf{x}_i)\big) \right)^2
$$
$$
\leqslant \mathcal{O}\big(LM(\tau)^2\big) \cdot L_S(\mathbf{W}^{(t)}). \quad \text{(C.6)}
$$

Then we can plug (C.5) and (C.6) into (C.1) and obtain

$$
\|\mathbf{W}^{(t)} - \mathbf{W}^*\|_F^2 - \|\mathbf{W}^{(t+1)} - \mathbf{W}^*\|_F^2
$$

$$\geqslant \frac{2\eta}{n} \sum_{i=1}^{n} \left[ (1 - 2\epsilon_{\mathrm{app}}(\tau))\ell\big(y_i f_{\mathbf{W}^{(t)}}(\mathbf{x}_i)\big) - \ell\big(y_i F_{\mathbf{W}^{(0)}, \mathbf{W}*}(\mathbf{x}_i)\big) \right] - \mathcal{O}\big(\eta^2 L M(\tau)^2\big) \cdot L_S(\mathbf{W}^{(t)})$$

$$\geqslant \left[ \frac{3}{2} - 4\epsilon_{\mathrm{app}}(\tau) \right] \eta L_S(\mathbf{W}^{(t)}) - \frac{2\eta}{n} \sum_{i=1}^{n} \ell\big(y_i F_{\mathbf{W}^{(0)}, \mathbf{W}*}(\mathbf{x}_i)\big),$$

where the last inequality is by $\eta = \mathcal{O}(L^{-1} M(\tau)^{-2})$ and merging the third term on the second line into the first term. Taking telescope sum from $t = 0$ to $t = t' - 1$ and plugging in the definition $\frac{1}{n} \sum_{i=1}^{n} \ell\big(y_i F_{\mathbf{W}^{(0)}, \mathbf{W}*}(\mathbf{x}_i)\big) = \epsilon_{\mathrm{NTRF}}$ completes the proof. $\qquad\square$

### C.2  PROOF OF LEMMA A.3

*Proof of Lemma A.3.* We first denote $\mathcal{W} = \mathcal{B}(\mathbf{W}^{(0)}, \widetilde{R} \cdot m^{-1/2})$, and define the corresponding neural network function class and surrogate loss function class as $\mathcal{F} = \{f_{\mathbf{W}}(\mathbf{x}) : \mathbf{W} \in \mathcal{W}\}$ and $\mathcal{G} = \{-\ell[y \cdot f_{\mathbf{W}}(\mathbf{x})] : \mathbf{W} \in \mathcal{W}\}$ respectively.

By standard uniform convergence results in terms of empirical Rademacher complexity (Bartlett and Mendelson, 2002; Mohri et al., 2018; Shalev-Shwartz and Ben-David, 2014), with probability at least $1 - \delta$ we have

$$\sup_{\mathbf{W} \in \mathcal{W}} |\mathcal{E}_S(\mathbf{W}) - \mathcal{E}_{\mathcal{D}}(\mathbf{W})| = \sup_{\mathbf{W} \in \mathcal{W}} \left| -\frac{1}{n} \sum_{i=1}^{n} \ell'\big[y_i \cdot f_{\mathbf{W}}(\boldsymbol{x}_i)\big] + \mathbb{E}_{(\mathbf{x}, y) \sim \mathcal{D}} \ell'\big[y \cdot f_{\mathbf{W}}(\mathbf{x})\big] \right|$$

$$\leqslant 2\widehat{\mathfrak{R}}_n(\mathcal{G}) + C_1 \sqrt{\frac{\log(1/\delta)}{n}},$$

where $C_1$ is an absolute constant, and

$$\widehat{\mathfrak{R}}_n(\mathcal{G}) = \mathbb{E}_{\xi_i \sim \mathrm{Unif}(\{\pm 1\})} \left\{ \sup_{\mathbf{W} \in \mathcal{W}} \frac{1}{n} \sum_{i=1}^{n} \xi_i \ell'\big[y_i \cdot f_{\mathbf{W}}(\boldsymbol{x}_i)\big] \right\}$$

is the empirical Rademacher complexity of the function class $\mathcal{G}$. We now provide two bounds on $\widehat{\mathfrak{R}}_n(\mathcal{G})$, whose combination gives the final result of Lemma A.3. First, by Corollary 5.35 in (Vershynin, 2010), with probability at least $1 - L \cdot \exp(-\Omega(m))$, $\|\mathbf{W}_l^{(0)}\|_2 \leqslant 3$ for all $l \in [L]$. Therefore for all $\mathbf{W} \in \mathcal{W}$, we have $\|\mathbf{W}_l\|_2 \leqslant 4$. Moreover, standard concentration inequalities on the norm of the first row of $\mathbf{W}_l^{(0)}$ also implies that $\|\mathbf{W}_l\|_2 \geqslant 0.5$ for all $\mathbf{W} \in \mathcal{W}$ and $l \in [L]$. Therefore, an adaptation of the bound in (Bartlett et al., 2017)[¶] gives

$$\widehat{\mathfrak{R}}_n(\mathcal{F}) \leqslant \tilde{\mathcal{O}} \left( \sup_{\mathbf{W} \in \mathcal{W}} \left\{ \frac{m^{1/2}}{\sqrt{n}} \cdot \left[ \prod_{l=1}^{L} \|\mathbf{W}_l\|_2 \right] \cdot \left[ \sum_{l=1}^{L} \frac{\|\mathbf{W}_l^\top - \mathbf{W}_l^{(0)\top}\|_{2,1}^{2/3}}{\|\mathbf{W}_l\|_2^{2/3}} \right]^{3/2} \right\} \right)$$

$$\leqslant \tilde{\mathcal{O}} \left( \sup_{\mathbf{W} \in \mathcal{W}} \left\{ \frac{4^L m^{1/2}}{\sqrt{n}} \cdot \left[ \sum_{l=1}^{L} (\sqrt{m} \cdot \|\mathbf{W}_l^\top - \mathbf{W}_l^{(0)\top}\|_F)^{2/3} \right]^{3/2} \right\} \right)$$

$$\leqslant \tilde{\mathcal{O}} \left( 4^L L^{3/2} \widetilde{R} \cdot \sqrt{\frac{m}{n}} \right). \tag{C.7}$$

We now derive the second bound on $\widehat{\mathfrak{R}}_n(\mathcal{G})$, which is inspired by the proof provided in (Cao and Gu, 2020). Since $y \in \{+1, 1\}$, $|\ell'(z)| \leqslant 1$ and $\ell'(z)$ is 1-Lipschitz continuous, by standard empirical Rademacher complexity bounds (Bartlett and Mendelson, 2002; Mohri et al., 2018; Shalev-Shwartz and Ben-David, 2014), we have

$$\widehat{\mathfrak{R}}_n(\mathcal{G}) \leqslant \widehat{\mathfrak{R}}_n(\mathcal{F}) = \mathbb{E}_{\xi_i \sim \mathrm{Unif}(\{\pm 1\})} \left[ \sup_{\mathbf{W} \in \mathcal{W}} \frac{1}{n} \sum_{i=1}^{n} \xi_i f_{\mathbf{W}}(\boldsymbol{x}_i) \right],$$

---

[¶]Bartlett et al. (2017) only proved the Rademacher complexity bound for the composition of the ramp loss and the neural network function. In our setting essentially the ramp loss is replaced with the $-\ell'(\cdot)$ function, which is bounded and 1-Lipschitz continuous. The proof in our setting is therefore exactly the same as the proof given in (Bartlett et al., 2017), and we can apply Theorem 3.3 and Lemma A.5 in (Bartlett et al., 2017) to obtain the desired bound we present here.

where $\widehat{\mathfrak{R}}_n(\mathcal{F})$ is the empirical Rademacher complexity of the function class $\mathcal{F}$. We have

$$\widehat{\mathfrak{R}}_n[\mathcal{F}] \leqslant \underbrace{\mathbb{E}_{\boldsymbol{\xi}}\left\{\sup_{\mathbf{W}\in\mathcal{W}} \frac{1}{n}\sum_{i=1}^{n} \xi_i\big[f_{\mathbf{W}}(\boldsymbol{x}_i) - F_{\mathbf{W}^{(0)},\mathbf{W}}(\boldsymbol{x}_i)\big]\right\}}_{I_1} + \underbrace{\mathbb{E}_{\boldsymbol{\xi}}\left\{\sup_{\mathbf{W}\in\mathcal{W}} \frac{1}{n}\sum_{i=1}^{n} \xi_i F_{\mathbf{W}^{(0)},\mathbf{W}}(\boldsymbol{x}_i)\right\}}_{I_2},$$

(C.8)

where $F_{\mathbf{W}^{(0)},\mathbf{W}}(\mathbf{x}) = f_{\mathbf{W}^{(0)}}(\mathbf{x}) + \big\langle\nabla_{\mathbf{W}} f_{\mathbf{W}^{(0)}}(\mathbf{x}), \mathbf{W} - \mathbf{W}^{(0)}\big\rangle$. For $I_1$, by Lemma 4.1 in (Cao and Gu, 2019), with probability at least $1 - \delta/2$ we have

$$I_1 \leqslant \max_{i\in[n]}\big|f_{\mathbf{W}}(\boldsymbol{x}_i) - F_{\mathbf{W}^{(0)},\mathbf{W}}(\boldsymbol{x}_i)\big| \leqslant \mathcal{O}\big(L^3\widetilde{R}^{4/3}m^{-1/6}\sqrt{\log(m)}\big),$$

For $I_2$, note that $\mathbb{E}_{\boldsymbol{\xi}}\big[\sup_{\mathbf{W}\in\mathcal{W}}\sum_{i=1}^{n}\xi_i f_{\mathbf{W}^{(0)}}(\boldsymbol{x}_i)\big] = 0$. By Cauchy-Schwarz inequality we have

$$I_2 = \frac{1}{n}\sum_{l=1}^{L}\mathbb{E}_{\boldsymbol{\xi}}\left\{\sup_{\|\widetilde{\mathbf{W}}_l\|_F\leqslant\widetilde{R}m^{-1/2}} \mathrm{Tr}\left[\widetilde{\mathbf{W}}_l^\top \sum_{i=1}^{n}\xi_i\nabla_{\mathbf{W}_l}f_{\mathbf{W}^{(0)}}(\boldsymbol{x}_i)\right]\right\}$$

$$\leqslant \frac{\widetilde{R}m^{-1/2}}{n}\sum_{l=1}^{L}\mathbb{E}_{\boldsymbol{\xi}}\left[\left\|\sum_{i=1}^{n}\xi_i\nabla_{\mathbf{W}_l}f_{\mathbf{W}^{(0)}}(\boldsymbol{x}_i)\right\|_F\right].$$

Therefore

$$I_2 \leqslant \frac{\widetilde{R}m^{-1/2}}{n}\sum_{l=1}^{L}\sqrt{\mathbb{E}_{\boldsymbol{\xi}}\left[\left\|\sum_{i=1}^{n}\xi_i\nabla_{\mathbf{W}_l}f_{\mathbf{W}^{(0)}}(\boldsymbol{x}_i)\right\|_F^2\right]}$$

$$= \frac{\widetilde{R}m^{-1/2}}{n}\sum_{l=1}^{L}\sqrt{\sum_{i=1}^{n}\big\|\nabla_{\mathbf{W}_l}f_{\mathbf{W}^{(0)}}(\boldsymbol{x}_i)\big\|_F^2}$$

$$\leqslant \mathcal{O}\left(\frac{L\cdot\widetilde{R}}{\sqrt{n}}\right),$$

where we apply Jensen's inequality to obtain the first inequality, and the last inequality follows by Lemma B.3 in (Cao and Gu, 2019). Combining the bounds of $I_1$ and $I_2$ gives

$$\widehat{\mathfrak{R}}_n[\mathcal{F}] \leqslant \widetilde{\mathcal{O}}\left(\frac{L\widetilde{R}}{\sqrt{n}} + \frac{L^3\widetilde{R}^{4/3}}{m^{1/6}}\right).$$

Further combining this bound with (C.7) and recaling $\delta$ completes the proof. $\square$

## C.3    PROOF OF LEMMA A.4

*Proof of Lemma A.4.* Different from the proof of Lemma 5.1, online SGD only queries one data to update the model parameters in each iteration, i.e., $\mathbf{W}^{i+1} = \mathbf{W}^i - \eta\nabla L_{i+1}(\mathbf{W}^{(i)})$. By this update rule, we have

$$\|\mathbf{W}^{(i)} - \mathbf{W}^*\|_F^2 - \|\mathbf{W}^{(i+1)} - \mathbf{W}^*\|_F^2$$

$$= 2\eta\langle\mathbf{W}^{(i)} - \mathbf{W}^*, \nabla_{\mathbf{W}}L_{i+1}(\mathbf{W}^{(i)})\rangle - \eta^2\sum_{l=1}^{L}\|\nabla_{\mathbf{W}_l}L_{i+1}(\mathbf{W}^{(i)})\|_F^2.$$

(C.9)

With exactly the same proof as (C.5) in the proof of Lemma 5.1, we have

$$\langle\mathbf{W}^{(t)} - \mathbf{W}^*, \nabla_{\mathbf{W}}L_i(\mathbf{W}^{(t)})\rangle \geqslant (1 - 2\epsilon_{\mathrm{app}}(\tau))\ell\big(y_i f_{\mathbf{W}^{(t)}}(\mathbf{x}_i)\big) - \ell\big(y_i F_{\mathbf{W}^{(0)},\mathbf{W}^*}(\mathbf{x}_i)\big), \quad \text{(C.10)}$$

for all $i = 0,\ldots,n'-1$. By the fact that $-\ell'(\cdot) \leqslant \ell(\cdot)$ and $-\ell'(\cdot) \leqslant 1$, we have

$$\sum_{l=1}^{L}\|\nabla_{\mathbf{W}_l}L_{i+1}(\mathbf{W}^{(i)})\|_F^2 \leqslant \sum_{l=1}^{L}\ell\big(y_{i+1}f_{\mathbf{W}_t}(\mathbf{x}_{i+1})\big)\cdot\|\nabla_{\mathbf{W}_l}f_{\mathbf{W}^{(i)}}(\mathbf{x}_{i+1})\|_F^2$$

$$\leqslant \mathcal{O}\big(LM(\tau)^2\big) \cdot L_{i+1}(\mathbf{W}^{(i)}). \tag{C.11}$$

Then plugging (C.10) and (C.11) into (C.9) gives

$$
\begin{aligned}
&\|\mathbf{W}^{(i)} - \mathbf{W}^*\|_F^2 - \|\mathbf{W}^{(i+1)} - \mathbf{W}^*\|_F^2 \\
&\quad\geqslant (2 - 4\epsilon_{\mathrm{app}}(\tau))\eta L_{i+1}(\mathbf{W}^{(i)}) - 2\eta\ell\big(y_i F_{\mathbf{W}^{(0)},\mathbf{W}*}(\mathbf{x}_i)\big) - \mathcal{O}\big(\eta^2 LM(\tau)^2\big)L_{i+1}(\mathbf{W}^{(i)}) \\
&\quad\geqslant (\frac{3}{2} - 4\epsilon_{\mathrm{app}}(\tau))\eta L_{i+1}(\mathbf{W}^{(i)}) - 2\eta\ell\big(y_i F_{\mathbf{W}^{(0)},\mathbf{W}*}(\mathbf{x}_i)\big),
\end{aligned}
$$

where the last inequality is by $\eta = \mathcal{O}(L^{-1}M(\tau)^{-2})$ and merging the third term on the second line into the first term. Taking telescope sum over $i = 0, \ldots, n' - 1$, we obtain

$$
\begin{aligned}
&\|\mathbf{W}^{(0)} - \mathbf{W}^*\|_F^2 - \|\mathbf{W}^{(n')} - \mathbf{W}^*\|_F^2 \\
&\quad\geqslant \Big(\frac{3}{2} - 4\epsilon_{\mathrm{app}}(\tau)\Big)\eta \sum_{i=1}^{n'} L_i(\mathbf{W}^{(i-1)}) - 2\eta \sum_{i=1}^{n'} \ell\big(y_i F_{\mathbf{W}^{(0)},\mathbf{W}*}(\mathbf{x}_i)\big). \\
&\quad\geqslant \Big(\frac{3}{2} - 4\epsilon_{\mathrm{app}}(\tau)\Big)\eta \sum_{i=1}^{n'} L_i(\mathbf{W}^{(i-1)}) - 2\eta \sum_{i=1}^{n} \ell\big(y_i F_{\mathbf{W}^{(0)},\mathbf{W}*}(\mathbf{x}_i)\big). \\
&\quad\geqslant \Big(\frac{3}{2} - 4\epsilon_{\mathrm{app}}(\tau)\Big)\eta \sum_{i=1}^{n'} L_i(\mathbf{W}^{(i-1)}) - 2n\eta\epsilon_{\mathrm{NTRF}}.
\end{aligned}
$$

This finishes the proof. □

## D   Experiments

In this section, we conduct some simple experiments to validate our theory. Since our paper mainly focuses on binary classification, we use a subset of the original CIFAR10 dataset (Krizhevsky et al., 2009), which only has two classes of images. We train a 5-layer fully-connected ReLU network on this binary classification dataset with different sample sizes ($n \in \{100, 200, 500, 1000, 2000, 5000, 10000\}$), and plot the minimal neural network width that is required to achieve zero training error in Figure 1 (solid line). We also plot $\mathcal{O}(n), \mathcal{O}(\log^3(n)), \mathcal{O}(\log^2(n))$ and $\mathcal{O}(\log(n))$ in dashed line for reference. It is evident that the required network width to achieve zero training error is polylogarithmic on the sample size $n$, which is consistent with our theory.

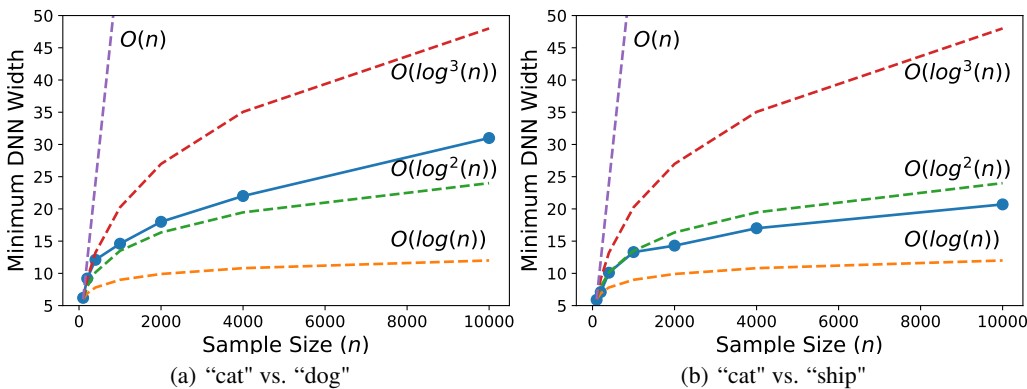

(a) "cat" vs. "dog"             (b) "cat" vs. "ship"

Figure 1: Minimum network width that is required to achieve zero training error with respect to the training sample size (blue solid line). The hidden constants in all $O(\cdot)$ notations are adjusted to ensure their plots (dashed lines) start from the same point.

