# OpenReview forum: "How Much Over-parameterization Is Sufficient to Learn Deep ReLU Networks?"
_ICLR.cc/2021/Conference — ICLR 2021 Poster_

### Official Review · AnonReviewer4 · 2020-10-21
**Well written paper with nice results but requiring precisions**

**Rating:** 8
**Confidence:** 5

**Review:**

The paper analyses the generalization properties of deep neural networks for classification tasks. The authors focus on the Neural Tangent Random Features (NTRF) class of functions to investigate these generalization properties.
In particular, the authors study the convergence of gradient descent (and its stochastic version). They establish that the training loss decreases until a certain level of the order of the best they can expect in the NTRF class. Then, the authors establish interesting generalization results with respect to the 0-1 loss.
This problem has been tackled recently in many papers. However, to establish similar results as those proved in this paper, the previous papers required a number of neurons per hidden layer to be polynomial with respect to the number of samples n. The authors shows here that a logarithmic (to a certain power) is actually sufficient when working with the NRTF class. Such results have been obtained recently in Ji and Telgarski (2020) but only for shallow network. In this paper, the results are generalized for deep neural networks.

The paper is clearly written and the comparison with the literature is complete and well-organized. The authors have worked hard to present, in a concise way and clearly their results. I appreciate it. The paper is fluid and nice to read.

Here are some remarks that might improve the presentation of the paper:

- It would be nice to recall the function $\sigma$ in Section 2

- In my opinion the remark:
" Moreover, while Ji and Telgarsky (2020)
essentially required all training data to be separable by a function in the NTRF function class with a
constant margin, our result does not require such data separation assumptions, and allows the NTRF
function class to misclassify a small proportion of the training data points",
is a big advantage of your analysis. Probably, you should spotlight it more clearly.

- Page 12 in the proof of Theorem 3.4. In the last inequality of the page, could you add that it holds because $-\ell'(x) \geq \ell(x)$ for all $x \in \mathbb R^d$.

- Proposition 4.2: you use two notations for the proportion of data not satisfying the margin assumption. You should replace all the $\sigma$ by $\rho$.

- Page 18: "where the equation follows from the fact that" --> $W^{0}$ should be $W^{t}$

- Before Equation (C.7). Could you add parenthesis around the product with  respect to $l$. Otherwise it is not very clear.

- Could you explain a bit more Assumption 4.1. Is it a strong assumption ? Do you have any insight wether it is verified in practice ?


To summarize, I enjoyed to review this paper. The results are clear and interesting. The paper deserves to be accepted for publication.

---

> ### Author Response · Authors · 2020-11-17
> **Response to Reviewer4**
>
> Thank you for your positive and valuable comments. We have addressed your suggestions and the typos you pointed out in the updated version.
>
>
> Q1: Allowing the NTRF function class to misclassify a small proportion of the training data points is a big advantage of your analysis.
>
> A1: Thanks for your supportive comments on our contribution, we have highlighted this in the introduction (third bullet of our contributions) of the revised version.
>
> Q2: Explain a bit more Assumption 4.1. Is it a strong assumption?
>
> A2: This is a good point and we believe Assumption 4.1 is not a strong assumption.  In fact, if the data can be well separated by NTK with a positive margin (as assumed in Cao and Gu, (2020), Ji and Telgarsky, (2020)), Assumption 4.1 also holds with proper choices of $m$ and $R$ (see the proof of Proposition 4.4 for more detail). Moreover, as it is widely known that NTK can learn the low-degree polynomials (Allen-Zhu et al., 2019a; Arora et al., 2019a), we can reason that if the training data can be separated by some polynomial functions with positive margin (e.g., data are linearly separable), NTK can also separate the training data and thus Assumption 4.1 holds as well.
> Besides, as you pointed out, Assumption 4.1 can also allow the NTRF function class to misclassify a small proportion of the training data points, which is not covered by previous works. This suggests that Assumption 4.1 can also handle noisy label setting, which is closer to the practice.
>
>
> Q3: Do you have any insight whether Assumption 4.1 is verified in practice ?
>
> A3: Since this assumption is on the linear separability of random features, it can be verified in practice by (i) generating the random features, (ii) solving a regularized linear classification problem over the random features, and (iii) calculating the classification error and margin on the correctly classified data. Then the obtained classification error and margin will correspond to the fraction $\rho$ and margin $m^{1/2}\gamma$ in Assumption 4.1 respectively.

---

### Official Review · AnonReviewer3 · 2020-10-28
**Overparametrisation Generalisation Results for Deeper Networks**

**Rating:** 6
**Confidence:** 2

**Review:**

The paper extends an existing proof for the sufficiency of polylogarithmic width for sharp learning guarantees of ReLU networks trained by (stochastic) gradient descent from shallow networks to deep networks. The theoretical analysis links the convergence of GD and SGD to the width of the network. The paper shows that polylogarithmic width is enough to give reasonable guarantees also for deep neural networks. It furthermore provides a generalisation bound in terms of network width.

The paper states a clearly formulated contribution and provides rigorous proofs for the claims stated. The analysis highly relies on the radius R of the NTRF function class and the authors provide a discussion showing the connection of the requirements on data separability to the values that R can take. The main lemma behind the proof and the sketch of the proof is also presented and explained.

The paper lacks a discussion of the applicability of the theory (e.g., how limiting are the assumptions of  the equal width layers, normalization by square root of m and the input norms being bounded by 1?), including a discussion on how the results could be generalized would be required. Also, since the paper extends a result on shallow networks to deep networks it should be discussed how depth affects the results. The generalization guarantees provided in this paper are vacuous without strict assumptions on data separability, as the authors state. However, it remains unclear whether the guarantee is non-vacuous for separable data. Could the authors argue whether the bound is non-vacuous for separable data?

But overall the contribution deems to be interesting still, so I suggest to accept the paper.

---

> ### Author Response · Authors · 2020-11-17
> **Response to Reviewer3**
>
> Thanks for encouraging and helpful comments.
>
> Q1: Applicability of the theory  (e.g., how limiting are the assumptions of the equal width layers, normalization by the square root of m, and the input norms being bounded by 1?)
>
> A1: These assumptions are made for the simplicity of presentation and theoretical analysis, which is commonly used in existing works (Allen-Zhu et al., 2019b, Zou et al., 2019, Du et al., 2019a). In fact, our theoretical results can be easily generalized to the settings with unequal width layers, as long as the smallest width satisfies our overparameterization condition.
>
> In addition, the input norm being bounded by 1 is also not essential. It can be relaxed to be bounded by some constant.
>
> In terms of the normalization by the square root of m, this is the standard initialization used in practice (e,g,. He initialization proposed in [1]). So our setting is consistent with the practice of deep learning.
>
> [1] He, K., Zhang, X., Ren, S. and Sun, J., 2015. Delving deep into rectifiers: Surpassing human-level performance on imagenet classification. In Proceedings of the IEEE international conference on computer vision (pp. 1026-1034).
>
>
> Q2: How depth affects the results?
>
> A2: For GD, Theorems 3.3 and 3.4 suggest that in order to guarantee sufficiently small test error (bounded by $\epsilon$), the neural network width may need to be polynomially large in the depth and the sample complexity may need to be exponentially large in the depth. This implies that when the depth increases, we may require a wider network and more training data to guarantee small test error. We also remark that a similar dependency on the depth $L$ has also appeared in previous works (Allen-Zhu et al., 2019b, Zou et al., 2019, Arora et al., 2019b).
>
> For SGD,  to achieve a small test error, according to Theorem 3.5, the width of deep networks and the sample complexity need to be polynomially large in the depth.
>
>
>
> Q3:  It remains unclear whether the guarantee is non-vacuous for separable data.
>
> A3: Our guarantee for separable data is non-vacuous. In specific, Proposition 4.2 and Proposition 4.4 imply that the generalization guarantee is non-vacuous if the training data are well separated (e.g., satisfying Assumption 4.1 or Assumption 4.3).  To see this, in Proposition 4.2 and Proposition 4.4, we have shown that an $\tilde{O}(1)$-wide network with $R = \tilde{O}(1)$ is sufficient to guarantee $\epsilon_{\text{NTRF}} \le \epsilon$. Plugging this into Theorems 3.4 and Theorem 3.5 yields $\tilde O(n^{-1/2})$ and $\tilde O(n^{-1})$ upper bounds on the test error respectively, which are non-vacuous.  We have updated the discussions after Propositions 4.2 and 4.4 in our revised version accordingly.

---

> > ### Comment · AnonReviewer3 · 2020-11-18
> > **Response**
> >
> > Thank you for clarifications and editions.
> >
> > A1 - I still do not quite connect the overall normalization of the neural network function by square root of m with the initialization techniques. Yes, at the first moment the network function can be written this way, but it does not mean that after training it will be constantly normalized.

---

> > > ### Author Response · Authors · 2020-11-18
> > > **Re Response**
> > >
> > > Thanks for your further question. We would like to clarify that we don’t require the output of the neural network to be constantly normalized after training. Since we are using logistic loss, the neural network after training will have a large margin on the training data, i.e., for each training data point, the output of the neural network function has the same sign as the label, and has a large magnitude. So the neural network function after training should not be constantly normalized.

---

### Official Review · AnonReviewer1 · 2020-10-28
**Nice result**

**Rating:** 7
**Confidence:** 2

**Review:**

- Overview

This paper greatly relaxed the rate of over-parametrization for deep neural networks.
Previously, shallow networks required fewer parameters (polylog(eps)), while deep networks required a large number of parameters (\Omega (eps^{-14})).
This paper shows that faster rates can be achieved in deep networks.
The key result is the analysis around the initial values using the Taylor approximation, which is independent of the shallowness of the layers.
This result allows the paper to achieve rates similar to existing shallow neural networks.

- Comment.

The results are impactful, showing that global convergence can be said to be possible with over-parametrization, even in deep neural networks with smaller number of nodes.
The paper is carefully written and the differences from existing research and the place of novelty are easy to see.

My question is what difference does this make to the smooth activation function case?
Lemma 5.1 doesn't seem to make much use of the ReLU property, is the result similar to the smooth case in this case?

Another question relates to optimality.
Is it possible to give an additional analysis of the theoretical limitations of the rates obtained here to see if they can be further improved?

The other one, which is not that important, is the value of 4^L that appears in Theorem 3.4.
This value will be a major obstacle to thinking about deep neural nets, but I wonder if it can be explained.

---

> ### Author Response · Authors · 2020-11-17
> **Response to Reviewer1**
>
> Thanks for recognizing the contribution of our work.
>
>
> Q1:  Is the result similar to the smooth case in this case?
>
> A1: Yes, the result for the smooth activation function will be similar. In specific, in order to extend our results from ReLU activation function to smooth activation functions, we need to revise Lemma A.1 to guarantee a relatively good linear approximation of the neural network around the initialization. For ReLU activation function, the linear approximation error can be bounded by $\tilde{O}(\tau^{4/3}m^{1/2})$, where $\tau$ is the radius of the neighborhood around initialization. For smooth activation function, the linear approximation error can be bounded by the second-order Taylor expansion and improved to be $\tilde{O}(\tau^{2}m^{1/2})$. Given the improved linear approximation error for the smooth activation function, the remaining proof will be almost the same.
>
>
>
>
> Q2: Optimality of sample complexity/iteration complexity/width requirement
>
> A2: In terms of sample complexity, our sample complexity of SGD is $O(1/\epsilon)$ which matches the faster rates for learning linear models. In this sense, we think our results may not be improvable. For GD, the sample complexity is $O(1/\epsilon^2)$ and is still improvable.
>
> In terms of iteration complexity, our iteration complexity results do not match the optimal complexity of the first-order optimization. We guess (S)GD with momentum may lead to better iteration complexity, while the analysis will be more involved.
>
> In terms of width requirements, since we only make polylogarithmic requirements, we believe our result is nearly optimal. A rigorous lower bound result could be an important future work.
>
>
>
>
> Q3: The value of 4^L in Theorem 3.4.
>
> A3: The 4^L factor is coming from uniform convergence based generalization bound (Rademacher complexity), which is adapted from Bartlett et al., (2017). In general, it is not clear whether this exponential dependence can be avoided. Nevertheless, in the neural tangent kernel (NTK) setting, this exponential dependence does not appear in the generalization error bound of SGD (See Theorem 3.5). So we guess this can also be improved for GD in the NTK setting.

---

### Official Review · AnonReviewer2 · 2020-10-28
**some useful but somewhat incremental improvements to finite-width NTK results**

**Rating:** 6
**Confidence:** 4

**Review:**

The paper studies optimization and generalization properties of deep relu networks trained with (stochastic) gradient descent on the logistic loss in the neural tangent kernel (NTK) regime. By using a new analysis that makes the "linearized" approximation as well as the L2 norm of the model in the approximate "random feature" kernel more explicit, the authors obtain results where the width only depends poly-logarithmically on the number of samples and 1/epsilon, for a test 0-1 loss of epsilon. This improves on previous analysis for deep networks, although it is similar to the two-layer result of Ji & Telgarsky.

I find the analysis interesting, more intuitive and practically relevant than previous studies, thanks to assumptions on the norm of the linearized model and the explicit bound on the approximation from linearization. The obtained guarantees are somewhat incremental are they are in large part similar to Ji & Telgarsky for the two-layer case, and remain in the NTK regime which only provides a limited picture of deep learning performance, but the extension to multiple layers and potentially other activations may be useful for the community, thus I remain on the accept side.

comments/questions:
- how do the dependencies of m and n on R or gamma compare to Ji & Telgarsky? more discussions on this comparisons at the end of sections 3.1 and 3.2 would be welcome
- the gap in sample complexity between GD and SGD is somewhat surprising (both the 1/eps^2 and the exponential dependency on L), is this just an artefact of the GD analysis in the generalization bound?
- in section 2, is the factor m^1/2 in the definition of f_W standard? how does this parameterization compare to previous works? (it seems like it may compensate the m^{-1/2} from the initialization of the first layer, which is often not present in other works, but this should be further discussed)
- in prop 4.4, it would be interesting to compare the obtained results to a simple linear model in the class NTRF, to see the cost of ensuring good linear approximation: does linear approximation require a larger m than what is needed just for good approximation of the infinite width kernel?

---

> ### Author Response · Authors · 2020-11-17
> **Response to Reviewer 2**
>
> Thank you for your detailed and useful comments. First of all, we would like to highlight the contribution of our work given Ji & Telgarsky. In specific, proving polylogarithmic width for deep networks is highly nontrivial since the proof technique proposed by Ji & Telgarsky highly relies on the fact that the one-hidden-layer ReLU network function is 1-homogeneous, which cannot be satisfied by deep neural networks.  Furthermore, our result is stronger by allowing the NTRF model to misclassify a proportion of training data. In comparison, Ji & Telgarsky requires the training data to be separable.
>
> Q1. Dependence of m and n on R and gamma, compared to Ji & Telgarsky
>
> A1. The dependency of width $m$ on $R$ in our paper is $\Omega(R^{8})$ (see Eq. (A.1) in the proof of Theorem 3.3). According to Proposition 4.4, we can prove that $R = \tilde{O}(1/\gamma)$ suffices under the same data assumption made in Ji & Telgarsky, leading to the overparameterization condition $m = \tilde O(1/\gamma^{8})$, which is the same as Ji & Telgarsky (see Corollary 3.3 in their paper). We have added a comment on it in Section 4.2 (right after Proposition 4.4) of the revised paper accordingly.
>
> Q2. The gap between the sample complexity of GD and SGD
>
> A2. We think the gap between sample complexities of GD and SGD is an artifact of the current GD analysis. The $O(1/\epsilon^2)$ sample complexity for GD is proved by uniform convergence-based generalization bound. In comparison, for SGD, by the online nature of the algorithm, we can apply a refined online-to-batch conversion argument (Ji and Telgarsky, 2020) to obtain a tighter $O(1/\epsilon)$ sample complexity. We believe a more careful analysis of GD using local Rademacher complexity can also lead to $O(1/\epsilon)$ sample complexity. We leave it as a future work as it will make the current analysis more involved. Regarding the exponential dependence on depth $L$ in GD, it is not clear how to improve it as this is inherited from the Rademacher complexity of deep neural networks as proved in Bartlett et al., (2017). We leave it as another future work direction.
>
> Q3. The parameterization compare to previous works
>
> A3. Our setting directly follows Cao and Gu (2019) and it is equivalent to other previous works. In specific, we initialize each entry in $\mathbf{W}_l^{\mathrm{ours}}$ as $N(0,2/m)$ and leave $m^{1/2}$ outside the last layer $\mathbf {W}_L^{\mathrm{ours}}$. The parameterization of previous works can be generally divided into two cases:
>
> Case 1: initialize weights in every layer with N(0,1) and add scaling factors 1/m^{1/2} before each hidden layer. Representative works include Jacot et al., (2018); Du et al., (2019a); Arora et al., (2019b).
> Case2: initialize weight in the hidden layer with N(0,2/m) and initialize weight in the last layer with N(0,1). Representative works include Allen-Zhu et al., (2019b); Zou et al., (2019).
>
> It is easy to show that, $\sqrt{m}\cdot \mathbf{W}_L^{\mathrm{ours}}$ is equivalent to $\mathbf{W}_L^{\mathrm{case1}}$ and $\mathbf{W}_L^{\mathrm{case2}}$. In addition, $\mathbf{W}_l^{\mathrm{ours}}$ is equivalent to $m^{-1/2}\mathbf{W}_l^{\mathrm{case1}}$ and $ \mathbf{W}_l^{\mathrm{case2}}$ for $l=1,\ldots,L-1$. Therefore all these parameterizations are essentially equivalent.
>
>
> Q4. Does linear approximation require a larger m than what is needed just for a good approximation of the infinite width kernel?
>
> A4. Yes, linear approximation requires a larger $m$ than what is needed for a good approximation of the infinite width kernel (in terms of data separation). To show this, from the proof of Theorem 3.3, by plugging the condition of $R$ in Proposition 4.4 into equation (A.1), we can obtain that the linear approximation requires $m$ to be $\tilde{O}(1/\gamma^{8})$. In contrast, according to Proposition 4.4, a good approximation of the infinite width kernel requires $m$ to be $\tilde{O}(1/\gamma^2)$. It is evident that linear approximation requires a larger $m$ than the approximation of infinite width kernel.

---

### Author Response · Authors · 2020-11-17
**Response to All Reviewers**

Thank you for your supportive and helpful reviews. We have addressed all your comments, and have updated the paper accordingly. The updates in the revised paper are highlighted in blue color.

---

### Decision · Program_Chairs · 2021-01-07
**Final Decision**

**Decision:**

Accept (Poster)

**Comment:**

The paper studies the convergence rate and generalization of deep ReLU networks trained with gradient descent and SGD in the NTK regime. Although the analysis technique is not really novel and heavily relies on past results, the paper is easy to follow and does provide some nice improvements compared to prior work (e.g. it require less overparametrization, and the NTRF function class is allowed to misclassify a fraction of the training data). Some of the results are very incremental, e.g. the generalization bound for GD seems to simply combine existing bounds on the Rademacher complexity from Bartlett et al. 2017 and from Cao et al. 2019. Nevertheless, the paper does have the potential to yield further improvements in the field and I therefore recommend acceptance as a poster.